# TET2-mediated tumor cGAS triggers endothelial STING activation to regulate vasculature remodeling and anti-tumor immunity in liver cancer

Hongwei Lv[1,2,3,8], Qianni Zong[1,2,8], Cian Chen[1,2,8], Guishuai Lv[1,2], Wei Xiang[3], Fuxue Xing[3], Guoqing Jiang[4], Bing Yan[4], Xiaoyan Sun[5], Yue Ma[3], Liang Wang[1,2], Zixin Wu[3], Xiuliang Cui[1,2], Hongyang Wang [1,2,3,6,7] & Wen Yang[1,2,3,6,7]

Induction of tumor vascular normalization is a crucial measure to enhance immunotherapy efficacy. cGAS-STING pathway is vital for anti-tumor immunity, but its role in tumor vasculature is unclear. Herein, using preclinical liver cancer models in *Cgas*/*Sting*-deficient male mice, we report that the interdependence between tumor cGAS and host STING mediates vascular normalization and anti-tumor immune response. Mechanistically, TET2 mediated IL-2/STAT5A signaling epigenetically upregulates tumor cGAS expression and produces cGAMP. Subsequently, cGAMP is transported via LRRC8C channels to activate STING in endothelial cells, enhancing recruitment and transendothelial migration of lymphocytes. In vivo studies in male mice also reveal that administration of vitamin C, a promising anti-cancer agent, stimulates TET2 activity, induces tumor vascular normalization and enhances the efficacy of anti-PD-L1 therapy alone or in combination with IL-2. Our findings elucidate a crosstalk between tumor and vascular endothelial cells in the tumor immune microenvironment, providing strategies to enhance the efficacy of combinational immunotherapy for liver cancer.

Liver cancer, primarily hepatocellular carcinoma (HCC), is one of the most frequent malignancies and the third leading causes of cancer-related death globally[1]. Currently, immune checkpoint blockade-based immunotherapies, particularly antibodies targeting programmed cell death-1 (PD-1)/programmed cell death ligand 1 (PD-L1) pathway have achieved remarkable success in the treatment of various malignancies including advanced HCC[2]. Nevertheless, anti-PD-1/PD-L1 monotherapy induces durable responses in only a subset of cancer patients and some responders relapse after a period of response[3]. Abnormal tumor blood vessels and the resulting hypoxia and immunosuppressive tumor microenvironment are key factors hindering immunotherapy efficacy, especially in liver cancer, which is highly angiogenic[4,5]. Anti-angiogenic therapy can not only normalize aberrant tumor blood vessels to improve intratumoral immune effector cell infiltration, but

---

[1]International Co-operation Laboratory on Signal Transduction, Eastern Hepatobiliary Surgery Hospital, Naval Medical University (Second Military Medical University), Shanghai 200438, China. [2]National Center for Liver Cancer, Naval Medical University (Second Military Medical University), Shanghai 201805, China. [3]Cancer Research Center, First Affiliated Hospital of USTC, Division of Life Sciences and Medicine, University of Science and Technology of China, Hefei, Anhui 230027, China. [4]Department of Hepatobiliary Surgery, Clinical Medical College, Yangzhou University, Yangzhou, Jiangsu 225000, China. [5]Hospital of Zhengzhou University, Zhengzhou, Henan 450000, China. [6]Shanghai Key Laboratory of Hepato-biliary Tumor Biology, Shanghai 200438, China. [7]Key Laboratory of Signaling Regulation and Targeting Therapy of Liver Cancer, Ministry of Education, Shanghai 200438, China. [8]These authors contributed equally: Hongwei Lv, Qianni Zong, Cian Chen. ✉e-mail: hywangk@vip.sina.com; woodeasy66@hotmail.com

also reverse the immunosuppressive microenvironment induced by angiogenic inducers, especially vascular endothelial growth factor (VEGF), providing rational for the combination of anti-angiogenic therapies with immune checkpoint blockade[6–8]. Recently, combination of anti-angiogenic therapy and immune checkpoint blockade have demonstrated efficacy in treating advanced HCC, making it the first-line therapy[8]. However, the vascular normalization 'time window' of current anti-angiogenic therapy is narrow and transient, limiting its enhancing effect on immunotherapy. Therefore, effective strategies to stably induce tumor vascular normalization and boost immunotherapy efficacy are urgently needed.

As a cytosolic DNA sensor, cyclic GMP-AMP synthase (cGAS) triggers innate immune responses through production of the second messenger cyclic GMP-AMP (cGAMP), which binds and activates the stimulator of interferon genes (STING)[9,10]. The activated STING recruits TANK binding kinase 1 (TBK1) and then phosphorylates interferon (IFN) regulatory factor 3 (IRF3) followed by induction of type I IFN and chemokines such as CCL5 and CXCL10[9,10]. Activated cGAS-STING pathway plays a vital role in anti-tumor immunity via T cell priming[11]. Yet, the regulation and mechanism of cGAS-STING pathway in cancer immunity remain to be fully understood. In addition to immune cell or tumor cell-intrinsic cGAS-STING activation, cGAS-STING pathway allows the cross-talk between tumor cells and surrounding non-tumor cells (or host cells) to regulate anti-tumor immunity[12]. The tumor-secreted cGAMP or DNA is transported to antigen presenting cells (APCs), such as dendritic cell (DC) and macrophage, and then activates the STING-induced type I IFN signaling pathway, finally triggering the anti-tumor immune response mediated by CD8+ T cells or NK cells[11,12]. Although DC is considered as the major cell type responding to tumor-derived cGAMP, many other cells in the tumor microenvironment with substantial STING expression, especially vascular endothelial cells, may also detect tumor-derived cGAMP. However, whether and how cGAS-STING pathway mediates the interaction between tumor cells and endothelial cells, as well as the role of cGAS-STING pathway in tumor vasculature remain largely unknown.

Intensive efforts have been invested to develop cGAS-STING agonists and several STING agonists have shown great promise in cancer immunotherapy in pre-clinical models[13,14]. Unfortunately, current STING agonists, most of which are synthetic analogs of cGAMP, are mainly delivered intratumorally due to poor bioavailability, limiting their clinical application and final therapeutic efficacy[13,14]. Therefore, major efforts are ongoing to develop more potent and selective cGAS-STING agonists to boost cancer immunotherapy.

Here, we describe the cross-talk between cGAS in tumor cells and STING in endothelial cells, which enhances transendothelial migration of lymphocytes, vascular normalization, and anti-tumor immunity. Based on the epigenetic regulation of tumor cGAS expression by ten-eleven translocation-2 (TET2) methylcytosine dioxygenase synergized with STAT5A signaling, we further stimulate TET2 activity using vitamin C (VC) to trigger tumor cGAS-cGAMP-endothelial STING pathway activation and induce tumor vascular normalization, thereby potentiating the therapeutic efficacy of immune checkpoint blockade in vivo.

## Results

### Tumor cGAS regulates vascular normalization and anti-tumor immune response in an intrinsic STING-independent manner

We first investigated the role of cancer cell-intrinsic cGAS and STING in tumor vascular normalization and anti-tumor immunity. The murine liver cancer cell line (Hepa1-6) with low endogenous Cgas and intact Sting expression was forcedly expressed with Cgas following depletion of its Sting via CRISPER-Cas9 or not to establish three cells of same origin with Cgas (−)/Sting (+), Cgas (−)/Sting (−), and Cgas (+)/Sting (−) expression (Fig. 1a, Supplementary Fig. 1a). Then, we challenged wild-type (WT) mice with these three cells respectively. Interestingly,

knockout of Sting in Hepa1-6 cells (Cgas (−)/Sting (−)) did not alter tumor growth compared to parental cells (Cgas (−)/Sting (+)) (Fig. 1b, c). Additionally, no obvious difference was observed in α-SMA+ pericyte coverage of CD31+ tumor vessels, which is one of the hallmarks of tumor vessel normalization[15], vascular permeability, or intratumoral T cell infiltration (Fig. 1d, e, Supplementary Fig. 2a–c), indicating that tumor-intrinsic STING expression is not required for vascular normalization and anti-tumor immunity under low tumor cGAS background. In contrast, on the background of Sting deficiency, Cgas overexpression in Hepa1-6 cells (Cgas (+)/Sting (−)) dramatically retarded tumor growth (Fig. 1b, c) and angiogenesis, increased pericyte coverage of tumor vessels, accompanied by mitigated vascular permeability and elevated intratumoral T cell infiltration (Fig. 1d, e, Supplementary Fig. 2a, b). Moreover, tumor vessels were generally more mature in Cgas over-expression tumors, as evidenced by the majority of the vessels expressing low levels of VEGF receptor 2 (VEGFR2), a molecular marker highly expressed by growing and immature vessels[16] (Fig. 1f, Supplementary Fig. 2c). In contrast, cancer cell-intrinsic Sting depletion exhibited no substantial effect on tumor blood vessel maturation. To further exclude the role of tumor-intrinsic STING under tumor-intrinsic cGAS overexpression background, we knocked out Sting in Cgas overexpression cells (Fig. 1g) and found that Sting deficiency (Cgas (+)/Sting (−)) did not alter tumor growth compared to parental cells (Cgas (+)/Sting (+)) (Fig. 1h, i). Moreover, no obvious difference was observed in α-SMA+ pericyte coverage of CD31+ tumor vessels or intratumoral T cell infiltration (Fig. 1j). Together, these results suggest that tumor cGAS controls vascular normalization and anti-tumor immune response in an intrinsic STING-independent manner.

To mimic a more faithful emulation of the endogenous cGAS/STING pathway levels observed in human HCC, a series of distinct clones with differential Cgas expression levels were selected (Fig. 1k). As shown in Fig. 1l, we observed a broader range of intracellular and extracellular cGAMP concentrations, spanning from a few hundred pg/mg protein to tens of thousands pg/mg protein. In vitro, the cell proliferation rates were similar after Cgas overexpression (Supplementary Fig. 1b). Importantly, we found that with higher Cgas expression in cancer cells, smaller tumor burdens were observed (Fig. 1m, n), while more pericyte coverage of tumor vessels and intratumoral T cell infiltration were detected (Fig. 1o), suggesting that tumor cGAS mediates tumor repression, vascular normalization, and anti-tumor immune response in a cGAS expression level-dependent manner. To further verify this phenomenon in human liver cancer, we established cGAS stably expressed human liver cancer cells (Huh7), which lack intrinsic cGAS and STING expressions, and forced cGAS expression did not alter Huh7 cells proliferation in culture (Supplementary Fig. 1c, d). Of note, cGAS cells-derived tumors in nude mice exhibited delayed growth (Supplementary Fig. 3a, b) and increased pericyte coverage of vessels, accompanied by reduced intratumoral hypoxia and more NK cells infiltration as compared to Ctrl cells-derived tumors (Supplementary Fig. 3c, d). Additionally, mouse-specific genes including downstream of STING activation IFNβ and ISGs, vascular stabilizing genes, endothelial-lymphocyte interaction-associated adhesion molecules were up-regulated in cGAS-proficient tumors (Supplementary Fig. 3e), indicating host STING activation and vascular normalization. In contrast, STING overexpression in cGAS cells (cGAS (+)/STING (+)) had little effect on tumor growth and vascular normalization compared with cGAS (+)/STING (−) cells (Supplementary Fig. 3f–i).

### Tumor cGAS and host STING mediates vascular normalization and anti-tumor immune response in an interdependence manner

We further assessed the roles of host STING in tumor cGAS-mediated vascular normalization and anti-tumor immune response. Cgas-proficient cells and ctrl cells were implanted in Sting-deficient (*Sting*−/−) mice or WT mice, respectively. We found that WT mice

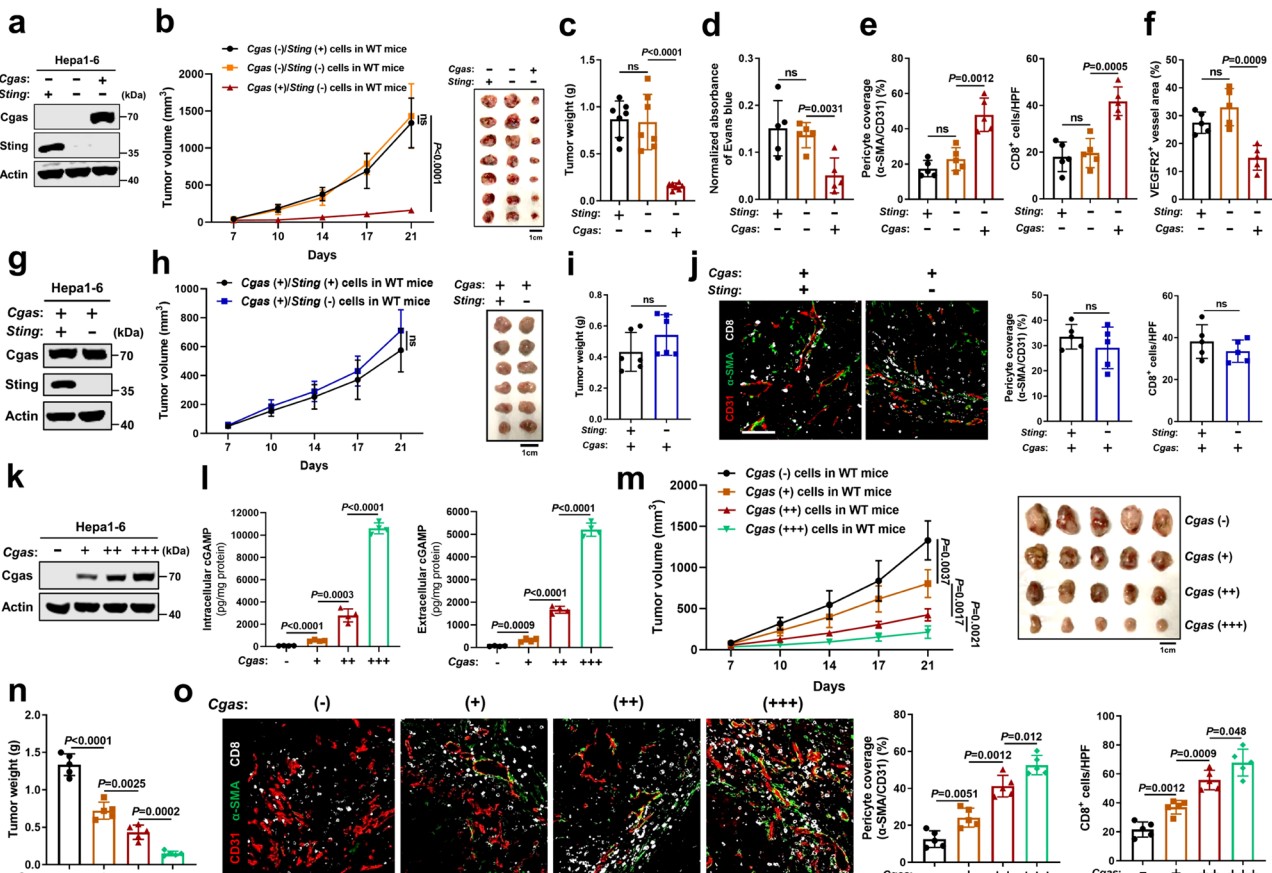

**Fig. 1 | Tumor cGAS regulates vascular normalization and anti-tumor immune response in an intrinsic STING-independent manner. a** Cgas and Sting protein levels in Hepa1-6 cells with *Sting* knockout following *Cgas* overexpression. Tumor growth curves (**b**) and tumor burdens (**c**) in WT mice injected subcutaneously with indicated cells for 3 weeks (*n* = 7 mice per group). **d** Normalized absorbance of Evans blue in indicated tumors (*n* = 5). **e** Quantification for pericyte (α-SMA⁺) coverage of tumor vessels (CD31⁺) and CD8⁺ T cells in indicated tumors (*n* = 5). HPF, high power field. **f** Quantification for VEGFR2⁺ tumor vessels (CD31⁺) in indicated tumors (*n* = 5). **g** Cgas and Sting protein levels in Hepa1-6 cells with *Cgas* over-expression following *Sting* knockout. Tumor growth curves (**h**) and tumor burdens (**i**) in WT mice injected subcutaneously with indicated cells for 3 weeks (*n* = 6 mice per group). **j** Representative immunofluorescence images and quantification for

pericyte (α-SMA⁺) coverage of tumor vessels (CD31⁺) and CD8⁺ T cells in indicated tumors (*n* = 5). HPF, high power field. Scale bars, 100 μm. **k** Cgas protein levels in Hepa1-6 cells with *Cgas* overexpression. **l** Intercellular and extracellular cGAMP levels from Hepa1-6 cells with *Cgas* overexpression (*n* = 4). Tumor growth curves (**m**) and tumor burdens (**n**) in WT mice injected subcutaneously with indicated cells for 3 weeks (*n* = 5 mice per group). **o** Representative immunofluorescence images and quantification for pericyte (α-SMA⁺) coverage of tumor vessels (CD31⁺) and CD8⁺ T cells in indicated tumors (*n* = 5). HPF, high power field. Scale bars, 100 μm. *P* values are calculated using two-way ANOVA (**b**, **h**, **m**), one-way ANOVA (**c–f**, **l**, **n**, **o**) and two-tailed unpaired Student's *t* test (**i**, **j**). ns not significant. Representative of *n* = 3 independent experiments (**a**, **g**, **k**). Source data are provided as a Source Data file.

implanted with Cgas-proficient cells developed significantly smaller tumors than WT mice injected with parental cells (Fig. 2a, b), while *Sting*⁻/⁻ mice implanted with Cgas-proficient cells developed comparable tumors than parental ctrl cells (Fig. 2c, d), suggesting that tumor cGAS-caused tumor repression depends on host STING. In contrast to Ctrl cells-derived tumors, a marked increase in pericyte coverage of tumor vessels was observed in Cgas-proficient tumors in WT mice (Fig. 2k). Furthermore, a significant reduction in intratumoral hypoxia, as shown by a hypoxia marker glucose transporter 1 (GLUT1) level[15], and a substantial increase in CD8⁺ T cells infiltration were detected in Cgas-proficient tumors versus Ctrl tumors in WT mice (Fig. 2k). Additionally, downstream of STING activation IFNβ and IFN-stimulated genes (ISGs), vascular stabilizing genes, and endothelial-lymphocyte interaction-associated adhesion molecules were up-regulated in Cgas-proficient tumors versus Ctrl tumors in WT mice (Supplementary Fig. 4a). Importantly, no obvious differences in tumor burdens, vascular normalization, CD8⁺ T cells infiltration were found in STING-deficient (*Sting*⁻/⁻) mice implanted with Cgas-proficient cells versus Ctrl cells (Fig. 2c, d, k, Supplementary Fig. 4b). Together, these results indicate that tumor cGAS-mediated

tumor repression, vascular normalization, and anti-tumor immune response rely on host STING.

We next challenged *Cgas*⁻/⁻ or WT mice with Cgas-proficient cells to address whether these tumor cGAS-mediated phenomenon are determined by host cGAS. Intriguingly, no significant difference in tumor growth was detected between *Cgas*⁻/⁻ and WT mice (Fig. 2e, f). Moreover, Cgas-deficient mice or WT mice challenged with Cgas-proficient cells showed comparable tumor vessel coverage by pericytes, intratumoral hypoxia, CD8⁺ T cell infiltration, and vascular normalizing genes expressions (Fig. 2k, Supplementary Fig. 4c). These findings suggest that host cGAS is not required for tumor cGAS-mediated mediated vascular normalization and anti-tumor immune response.

To test whether host STING relies on tumor cGAS to regulate vascular normalization and anti-tumor immune response, we challenged *Sting*⁻/⁻ mice and WT mice with Cgas-proficient cells or ctrl cells, respectively. Ctrl cells developed similar tumor burdens in *Sting*⁻/⁻ mice versus WT mice (Fig. 2g, h), whereas Cgas-proficient cells exhibited faster growth in *Sting*⁻/⁻ mice versus WT mice (Fig. 2i, j). In contrast to Ctrl cells-derived tumors, a marked reduction in

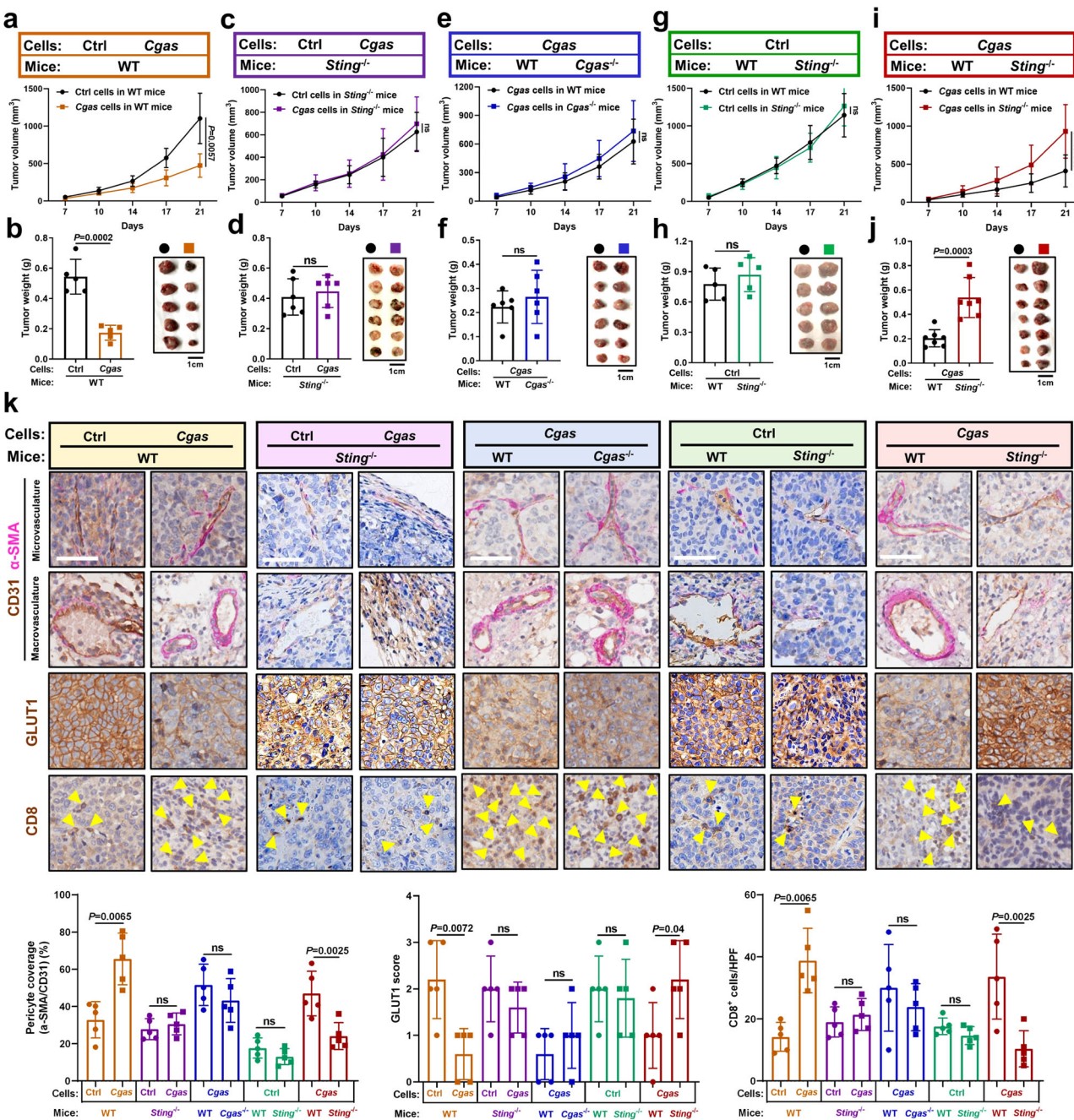

**Fig. 2 | Tumor cGAS and host STING mediates vascular normalization and anti-tumor immune response in an interdependence manner.** Tumor growth curves (**a**) and tumor burdens (**b**) in WT mice injected subcutaneously with Hepa1-6-*Cgas* or Ctrl cells for 3 weeks (*n* = 5 mice per group). Tumor growth curves (**c**) and tumor burdens (**d**) in *Sting*$^{-/-}$ mice injected subcutaneously with Hepa1-6-*Cgas* or Ctrl cells for 3 weeks (*n* = 6 mice per group). Tumor growth curves (**e**) and tumor burdens (**f**) in *Cgas*$^{-/-}$ or WT mice injected subcutaneously with Hepa1-6-*Cgas* cells for 3 weeks (*n* = 6 mice per group). **g**, **h** Tumor growth curves (**g**) and tumor burdens (**h**) in *Sting*$^{-/-}$ or WT mice injected subcutaneously with Hepa1-6-Ctrl cells for 3 weeks

(*n* = 5 mice per group). Tumor growth curves (**i**) and tumor burdens (**j**) in *Sting*$^{-/-}$ or WT mice injected subcutaneously with Hepa1-6-*Cgas* cells for 3 weeks (*n* = 7 mice per group). **k** Representative immunohistochemical images and quantification for pericyte (α-SMA$^+$) coverage of vessels (CD31$^+$), GLUT1$^+$ hypoxic area, and CD8$^+$ T cells in indicated tumors (*n* = 5). The yellow arrows represent CD8$^+$ T cells. Scale bars, 100 μm. *P* values are calculated using two-way ANOVA (**a**, **c**, **e**, **g**, **i**) and two-tailed unpaired Student's *t* test (**b**, **d**, **f**, **h**, **j**, **k**). ns, not significant. Source data are provided as a Source Data file.

pericyte coverage of tumor vessels and CD8$^+$ T cells infiltration, but a significant increase in intratumoral hypoxia were observed in Cgas-proficient tumors from *Sting*$^{-/-}$ mice versus WT mice (Fig. 2k). Additionally, no obvious differences in downstream of STING activation IFNβ and ISGs, vascular stabilizing genes, and endothelial-lymphocyte interaction-associated adhesion molecules were observed in ctrl tumors from *Sting*$^{-/-}$ mice versus WT mice, but these

genes were down-regulated in Cgas-proficient tumors from *Sting*$^{-/-}$ mice versus WT mice (Supplementary Fig. 4d, e). These findings demonstrate that host STING-mediated tumor repression, vascular normalization, and anti-tumor immune response depend on tumor cGAS. Collectively, these findings indicate that tumor cGAS and host STING mediates vascular normalization and anti-tumor immune response in an interdependence manner.

## Tumor cGAS produces cGAMP to activate endothelial STING and promote lymphocyte trafficking

To further explore the crosstalk of cGAS-STING between tumor cells and host cells, we first assessed cGAS and STING expression in tumor microenvironment. In Human Protein Atlas (HPA), cGAS was positive expressed in liver cancer cells from most patients, but positive STING expression in cancer cells was only detected in one of 12 patients (Supplementary Fig. 5a). Similarly, 87.5% of the 103 HCC patients from our hospital (EHBH) showed positive expression of cGAS in cancer cells, while only less than 9% of the patients displayed positive STING expression in cancer cells (Supplementary Fig. 5b). In contrast, most distinct STING expression was identified in endothelial cells compared with other host cells in human liver tumor microenvironment by single cell type clusters[17] (Supplementary Fig. 5c). Furthermore, STING expression in CD31[+] tumor vessels was verified in serial sections of HCC samples from the same patient (Supplementary Fig. 5d). Therefore, these results suggest that liver cancer cells express cGAS, but hardly express STING, whereas endothelial cells show most distinct STING expression and they are most important target of cGAMP, as they are more abundant than other host cells in tumor microenvironment. Additionally, although T cell is an important contributor to tumor vessel normalization[7], tumor cGAS was sufficient to repress tumor angiogenesis and induce vascular normalization in immunodeficient mice (Supplementary Fig. 4a–e), excluding the essential role of T cell-intrinsic STING in tumor cGAS-mediated tumor vasculature remodeling. Together, these data support the preferential crosstalk between tumor cGAS and endothelial STING in live caner, which is highly angiogenic.

Gene set enrichment analysis (GSEA) (Panther pathway analysis) revealed that tumor cGAS was highly interrelated with both T cell activation and angiogenesis in liver cancer from TCGA (Fig. 3a), indicating the regulatory role of tumor cGAS in vasculature remodeling. We further assessed the correlation between tumor cGAS and endothelial STING activation in human liver cancer. CD31[+] tumor-associated endothelial cells (TECs) and the remaining non-TEC cells (mainly tumor parenchymal cells) were sorted from liver cancer tissues of the same patient. Importantly, cGAS expression in non-TEC cells was positively correlated with STING and ISGs expressions in paired TECs (Fig. 3b). Moreover, compared to the matched TECs, CD31[+] normal endothelial cells (NECs) expressed increased STING and ISGs (Fig. 3c), confirming the role of endothelial STING in vascular normalization. Next, we verified the activation of endothelial STING by liver cancer cell cGAS through in vitro experiments. After confirming that Cgas overexpression boosted cGAMP secretion in liver cancer cells by detecting the levels of cGAMP in intracellular and extracellular media (Fig. 3d), we then tested whether cGAMP produced by liver cancer cells could activate STING in surrounding endothelial cells, using the formation of perinuclear STING aggregates as readout for STING activation by cGAMP[10,18]. The formation of Sting aggregates upon cGAMP stimulation was first confirmed in endothelial cells expressing *Sting*-Cherry plasmid following endogenous *Sting* depletion (Supplementary Fig. 6a, b). Then, *Sting*-Cherry-expressed endothelial cells were co-cultured with GFP-labeled Cgas and GFP-labeled Ctrl cells, respectively. We found that Sting aggregates were much more prominent in endothelial cells co-cultured with Cgas cells versus Ctrl cells (Fig. 3e). Additionally, compared with Ctrl cells-derived conditioned medium (CM), stimulation of endothelial cells by Cgas cell-derived CM (much more cGAMP in CM form Cgas cells versus Ctrl cells were confirmed in Fig. 3d) led to Sting activation, as shown by elevated phosphorylation of Sting and Tbk1, as well as increased Ifnβ and ISGs expressions (Fig. 3f, g).

We next investigated the role of endothelial STING activation by liver cancer cells in endothelial cells functionalities. Direct activation of Sting pathway by cGAMP (cGAMP-induced STING pathway activation was confirmed in Supplementary Fig. 6c, d) inhibited endothelial cell proliferation and their tube-like structure formation ability in the Matrigel assay (Supplementary Fig. 6e, g). Similarly, Cgas cell-derived CM treatment also partially repressed endothelial cell proliferation and tube formation (Supplementary Fig. 6f, h). Additionally, the migratory characteristics of endothelial cells were suppressed after co-culture with Cgas cells, as shown in the tumor-chemotaxis assay (Fig. 4a). Since tumor vascular normalization improves lymphocyte infiltration into tumors[6], we evaluated whether STING-activated endothelial cells supported more lymphocytes infiltration. By utilizing a modified lymphocytes transendothelial migration assay as described previously[19], we found that more lymphocytes transmigrated through the cGAMP-treated endothelial cell barrier versus Ctrl barrier (Fig. 4b) or endothelial cell barrier co-cultured with Cgas cells versus Ctrl cells (Fig. 4c). However, Sting-deficient lymphocytes and WT lymphocytes did not display any difference in transendothelial migration (Fig. 4d, e). Therefore, high cGAS expressed cancer cells enhances lymphocytes transendothelial migration by activation of endothelial STING, rather than lymphocytes STING pathway, through cGAMP secretion.

Mechanistically, endothelial cells exhibited increased vascular endothelial (VE)-cadherin (VE-Cad) expression following cGAMP treatment (Fig. 4f) or mixed culture with Cgas cells versus Ctrl cells (Fig. 4g), suggesting that Cgas cell-induced endothelial STING activation maintains the stability of endothelial cell junctions, which are crucial for leukocyte trafficking[20]. Additionally, after Sting activation by cGAMP or Cgas cell-derived CM stimulation, endothelial cells displayed increased expression of adhesion molecules involved in endothelial-lymphocyte interaction including Icam, Vcam, E-selectin (Sele), and L-selectin (Sell) (Fig. 4h, i). In contrast, Sting deficiency impaired these adhesion molecules expressions (Fig. 4j). Furthermore, blocking the IFNβ signaling pathway by STAT1 activation inhibitor fludarabine (Flura) reversed cGAMP-induced Ccl5 expression, but not Icam expression (Supplementary Fig. 6i), and IFNβ treatment increased Ccl5 expression, but has no obvious effect on Icam expression in endothelial cells (Supplementary Fig. 6j), suggesting the regulation of endothelial-lymphocyte interaction-associated adhesion molecules mainly by STING pathway, rather than its target genes IFNβ and ISGs. Collectively, these data indicate that tumor cGAS produces cGAMP to activate STING pathway in endothelial cells, further suppressing angiogenesis and promoting lymphocyte trafficking via maintenance of endothelial cell junction stability and upregulation of endothelial-lymphocyte interaction-associated adhesion molecules.

## Tumor-derived cGAMP transport via LRRC8C channels to activate STING in endothelial cells

We next investigate how cGAMP produced by cGAS in liver cancer cells is transported to endothelial cells. In addition to the gap junction, which directly connects adjacent cells[21], recent studies have reported that cGAMP is transmitted between cells through solute carriers (SLCs) including SLC19A1 and SLC46A2, and volume-regulated anion channels (VRACs), formed by LRRC8 heteromers[22–25]. Since Cgas cells-derived CM was sufficient to activate endothelial STING (Fig. 3e–g), gap junctions were excluded. Interestingly, among above reported SLCs and VRACs, tumor LRRC8C expression was most positively correlated with cGAS-STING and vascular normalization-associated genes including vascular stabilization, endothelial-lymphocyte interaction, pericytes and T cell chemotaxis in TCGA database (Fig. 5a). Additionally, the strongest positive correlation was found between tumor LRRC8C expression and CD8[+] T cell infiltration (Fig. 5b). Among LRRC8A-E, high tumor LRRC8C expression predicted better prognosis in liver cancer patients (Supplementary Fig. 7a–e). Similar to tumor cGAS, GSEA (Panther pathway analysis) showed that tumor LRRC8C was highly interrelated with both T cell activation and angiogenesis in liver cancer from TCGA dataset (Fig. 5c). Therefore, these results by bioinformatic analysis imply that LRRC8C may mediate the

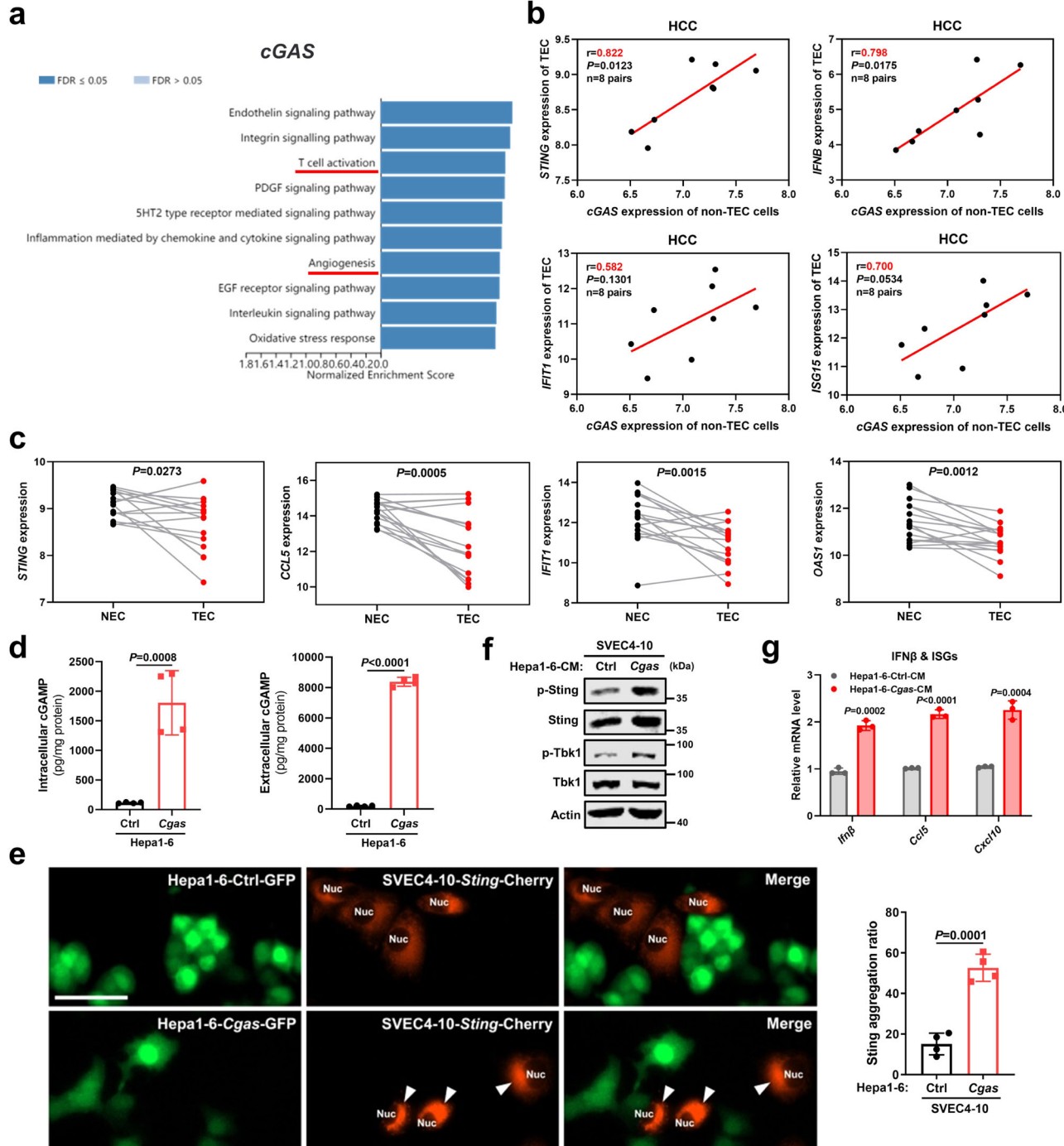

**Fig. 3 | Tumor cGAS produces cGAMP to activate STING in endothelial cells.**
**a** GSEA (Panther pathway analysis) of cGAS in liver cancer from TCGA ($n = 371$).
**b** Correlation analysis of cGAS expression in non-tumor-associated endothelial cell (non-TEC) cells with STING and ISGs expressions in paired TECs from liver cancer tissues of the same patient via magnetic-activated cell sorting (MACS) in the GSE51401 dataset ($n = 8$ pairs). **c** STING and ISGs expressions of TECs versus normal endothelial cells (NECs) in the GSE51401 dataset ($n = 16$ pairs). **d** Intercellular and extracellular cGAMP levels from Hepa1-6-*Cgas* or Ctrl cells ($n = 4$). **e** Representative immunofluorescence images and quantification for Sting aggregates in SVEC4-10

cells with *Sting*-Cherry overexpression co-cultured with GFP-labeled Hepa1-6-*Cgas* or Ctrl cells ($n = 4$). The white arrows represent Sting aggregates in SVEC4-10 cells. Scale bars, 50 µm. **f** Protein levels of markers in the Sting pathway in SVEC4-10 cells after exposure to CM from Hepa1-6-*Cgas* or Ctrl cells. Representative of $n = 3$ independent experiments. **g** mRNA levels of Ifnβ and ISGs in SVEC4-10 cells after exposure to CM from Hepa1-6-*Cgas* or Ctrl cells ($n = 3$). $P$ values are calculated using two-tailed Pearson correlation coefficient (**b**), two-tailed paired Student's $t$ test (**c**) and unpaired Student's $t$ test (**d**, **e**, **g**). Source data are provided as a Source Data file.

---

transmission of cGAMP between liver cancer cells and vascular endothelial cells.

To further test the role of LRRC8C in cGAMP transportation, Lrrc8c was knocked down in Cgas cells (Fig. 5d). It was found that the intracellular level of cGAMP was not obvious affected, while the

extracellular level of cGAMP was significantly reduced (Fig. 5e), indicating that LRRC8 mediates the secretion of cGAMP in liver cancer cells. Furthermore, endothelial Sting activation by Cgas cells-derived CM was dramatically weakened after Lrrc8c knockdown, as shown by decreased phosphorylation of Sting and Tbk1, as well as Ifnβ and ISGs

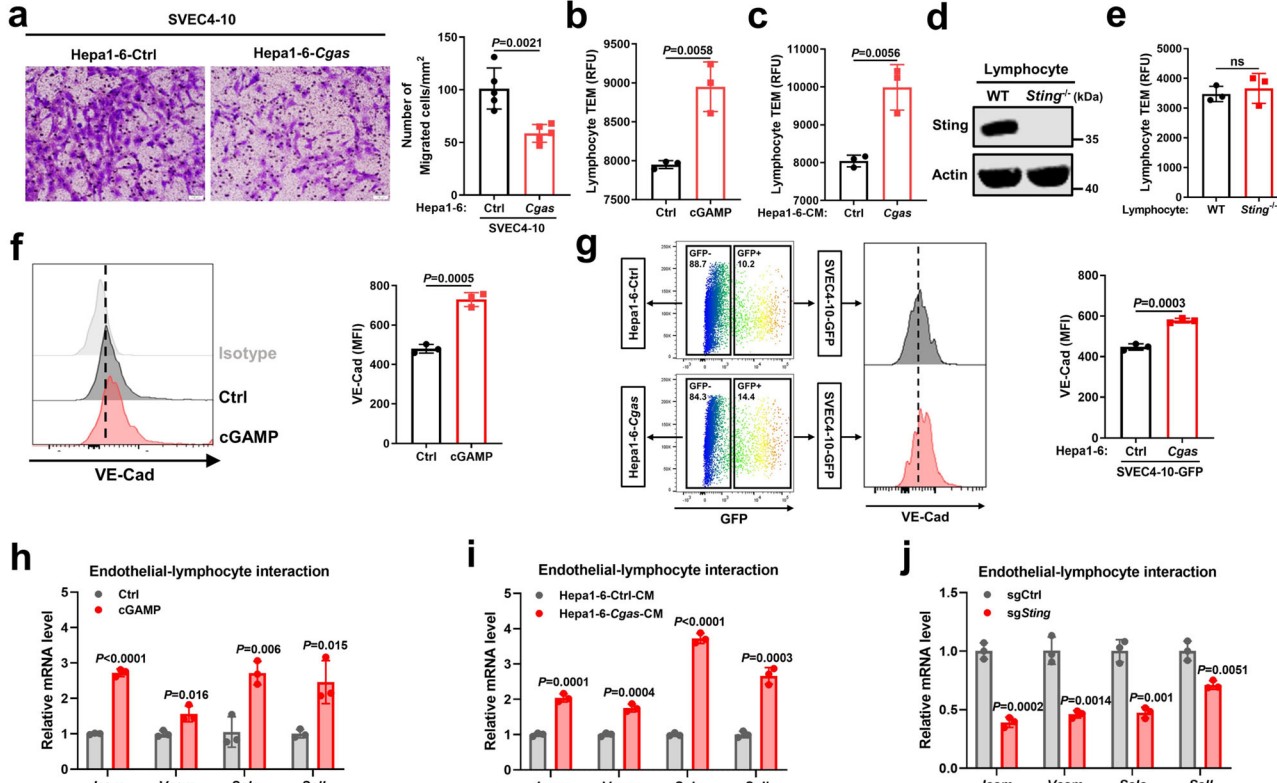

**Fig. 4 | Tumor cGAS-induced endothelial STING activation promotes lymphocyte trafficking. a** Representative images and quantifications of SVEC4-10 cells that migrated towards Hepa1-6-*Cgas* or Ctrl cells in Transwell assays (*n* = 5). **b** Transendothelial migration (TEM) of mouse splenic lymphocytes through SVEC4-10 cell barrier with cGAMP pre-treatment or not (*n* = 3). **c** TEM of mouse splenic lymphocytes through SVEC4-10 cell barrier with pre-treatment of CM from Hepa1-6-*Cgas* or Ctrl cells (*n* = 3). **d** Protein levels of Sting in splenic lymphocytes from *Sting*−/− or WT mice. Representative of *n* = 3 independent experiments. **e** TEM of *Sting*−/− or WT mice-derived lymphocytes through SVEC4-10 cell barrier (*n* = 3).

**f, g** Flow cytometric analysis and mean fluorescence intensity (MFI) of surface VE-Cad in SVEC4-10 cells after cGAMP treatment (**f**) (*n* = 3), co-cultured with Hepa1-6-*Cgas* or Ctrl cells (**g**) (*n* = 3). **h** mRNA levels of indicated genes in SVEC4-10 cells after cGAMP treatment (*n* = 3). **i** mRNA levels of indicated genes in SVEC4-10 cells co-cultured with Hepa1-6-*Cgas* or Ctrl cells (*n* = 3). **j** mRNA levels of indicated genes in SVEC4-10 cells with *Sting* knocked out (sg*Sting*) or Ctrl cells (sgCtrl) (*n* = 3). *P* values are calculated using unpaired Student's *t* test (**a**–**c**, **e**–**j**). ns, not significant. Source data are provided as a Source Data file.

expressions (Fig. 5f, g), together suggesting that cGAMP produced by liver cancer cells is secreted out of the cell through LRRC8C channels, thereby activating STING of endothelial cells. Since LRRC8 channels have been proved to be a bidirectional cGAMP transporter[24,25], we next determined whether cGAMP secreted by cancer cells enters endothelial cells also through LRRC8C. When Lrrc8c expression was blocked in endothelial cells, either cGAMP (Fig. 5h, i) or Cgas cells-derived CM (Fig. 5j, k) failed to activate endothelial Sting pathway, as evidenced by reduced phosphorylation of Sting and Tbk1, as well as decreased Ifnβ and ISGs expressions, indicating that endothelial cells use LRRC8C channels to import cGAMP, similar to previous reports[24]. Furthermore, knockdown of Lrrc8c in Cgas cells or endothelial cells also significantly repressed cGAMP or Cgas cells-derived CM induced endothelial-lymphocyte interaction-associated adhesion molecules expressions in endothelial cells (Fig. 5l–n). Together, these data indicate that LRRC8C channels mediate cGAMP transmission from liver cancer cells to vascular endothelial cells, thereby activating endothelial STING pathway.

## TET2 synergizes with IL-2/STAT5A signaling to promote tumor cGAS expression and cGAMP secretion

Although cGAS was positive expressed in HCC, its cGAS expression at transcription level was the lowest among pan-cancer, accompanied by the highest methylation of cGAS gene promoter[26] (Fig. 6a). Indeed, cGAS expression levels were negatively correlated with their methylation levels (*r* = −0.51, *P* < 0.0001) in HCC from TCGA dataset (Fig. 6b). DNA demethylation is mainly controlled by TET dioxygenases, which

are capable of converting 5-methylcytosine (5mC) to 5-hydroxymethylcytosine (5hmC)[27]. Interestingly, among TET family members, tumor suppressor TET2 expression was most positively correlated with cGAS expression and no negative correlation between tumor cGAS expression and DNA methyltransferases (DNMTs) was observed in HCC from TCGA and ICGC datasets (Fig. 6c). To further investigate whether TET2 regulates cGAS expression, we overexpressed Tet1, Tet2, and Tet3 in liver cancer cells, respectively. Notably, Tet2 overexpression significantly increased Cgas expression in comparison to Tet1 and Tet3, at both the transcript and protein levels (Fig. 6d, e). Furthermore, Tet2 overexpression also enhanced Cgas enzyme activity, as supported by the elevated intracellular and extracellular levels of cGAMP (Fig. 6f). Together, these findings suggest that TET2 is a major mediator of cGAS expression in liver cancer.

Because TET family often interacts with other signaling, especially JAK/STAT pathway, to synergistically mediate target gene expression[28], we questioned whether TET2 cooperated with STAT signaling to regulate cGAS expression. Surprisingly, among STAT family members (STAT1-6), STAT5A showed a most positive correlation with cGAS in the high TET2 expression group rather than in low TET2 expression group from HCC ICGC datasets (Fig. 6g). In vitro, disruption of Stat5a, but not Stat1, which exhibited a positive correlation with cGAS regardless of TET2 expression (Fig. 6g), impaired Cgas expression in liver cancer cells (Fig. 6h, i). Conversely, activation of Stat5a upon IL-2 stimulation upregulated Cgas and this IL-2-induced Cgas was reversed after Stat5a knockdown (Fig. 6j), indicating the

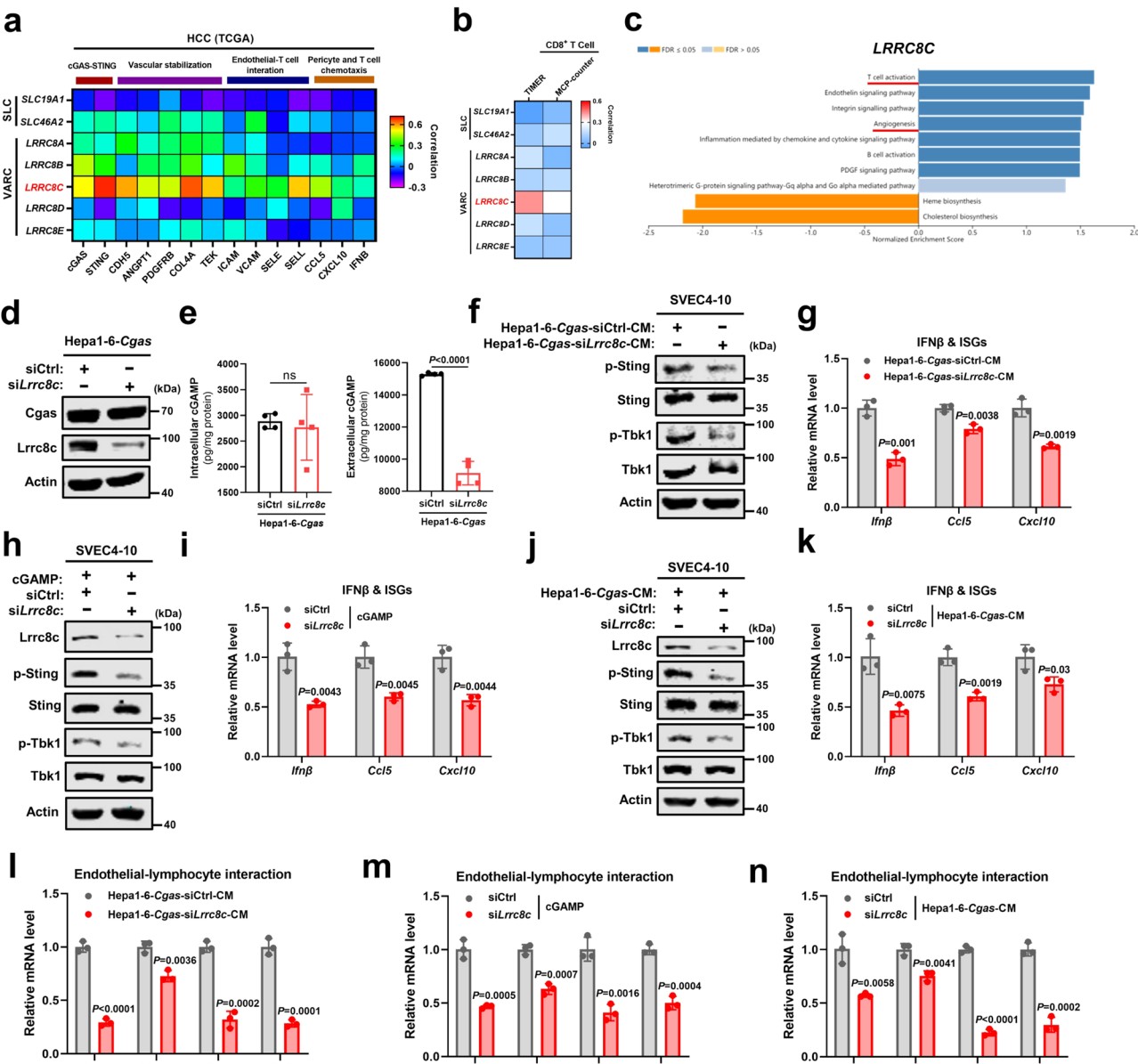

**Fig. 5 | Tumor-derived cGAMP transport via LRRC8C channel to activate STING in endothelial cells.** **a** Pearson's correlation of SLC (SLC19A1/SLC46A2) and VARC (LRRC8A-E) expressions with genes related to cGAS-STING, vascular stabilization, endothelial-T cell interaction, and pericyte/T cell chemotaxis in HCC from TCGA dataset (n = 371). **b** Pearson's correlation of SLC (SLC19A1/SLC46A2) and VARC (LRRC8A-E) expressions with CD8[+] T cell infiltration in HCC from TCGA dataset through TIMER and MCP-counter (n = 371). **c** GSEA (Panther pathway analysis) of LRRC8C in liver cancer from TCGA (n = 371). **d** Protein levels of Cgas and Lrrc8c in Hepa1-6-*Cgas* cells with *Lrrc8c* knocked down by siRNA (Hepa1-6-*Cgas*-si*Lrrc8c*) or Ctrl cells (Hepa1-6-*Cgas*-siCtrl). **e** Intercellular and extracellular cGAMP levels from Hepa1-6-*Cgas*-si*Lrrc8c* or *Cgas*-siCtrl cells (n = 4). **f** Protein levels of markers in the Sting pathway in SVEC4-10 cells after exposure to CM from Hepa1-6-*Cgas*-si*Lrrc8c* or *Cgas*-siCtrl cells. **g** mRNA levels of Ifnβ and ISGs in SVEC4-10 cells after exposure to CM from Hepa1-6-*Cgas*-si*Lrrc8c* or *Cgas*-siCtrl cells (n = 3). **h** Protein levels of Lrrc8c and markers in the Sting pathway in SVEC4-10-si*Lrrc8c* or siCtrl cells after cGAMP treatment. **i** mRNA levels of Ifnβ and ISGs in SVEC4-10-si*Lrrc8c* or siCtrl cells after cGAMP treatment (n = 3). **j** Protein levels of Lrrc8c and markers in the Sting pathway in SVEC4-10-si*Lrrc8c* or siCtrl cells after exposure to CM from Hepa1-6-*Cgas* cells. **k** mRNA levels of Ifnβ and ISGs in SVEC4-10-si*Lrrc8c* or siCtrl cells after exposure to CM from Hepa1-6-*Cgas* cells (n = 3). **l** mRNA levels of indicated genes in SVEC4-10 cells after exposure to CM from Hepa1-6-*Cgas*-si*Lrrc8c* or *Cgas*-siCtrl cells (n = 3). **m** mRNA levels of indicated genes in SVEC4-10-si*Lrrc8c* or siCtrl cells after cGAMP treatment (n = 3). **n** mRNA levels of indicated genes in SVEC4-10-si*Lrrc8c* or siCtrl cells after exposure to CM from Hepa1-6-*Cgas* cells (n = 3). *P* values are calculated using two-tailed unpaired Student's *t* test (**e**, **g**, **i**, **k**–**n**). ns, not significant. Representative of n = 3 independent experiments (**d**, **f**, **h**, **j**). Source data are provided as a Source Data file.

regulatory role of IL-2/STAT5A signaling in cGAS expression. However, IL-2 treatment failed to induce Cgas expression after Tet2 deficiency (Fig. 6k). In contrast, Tet2 overexpression in combination with IL-2 stimulation led to more increase in Cgas expression than either alone (Fig. 6l), suggesting that TET2 synergizes with STAT5A signaling to promote tumor cGAS expression. Mechanistically, Tet2 interacted with activated Stat5a (p-Stat5a) and this interaction was stronger upon

IL-2 stimulation (Fig. 6m). Furthermore, Tet2 bound to the Cgas promoter and this binding was significantly enhanced in the presence of IL-2 (Fig. 6n). To further verify the role of Tet2 in Cgas demethylation, Tet2 overexpression was sufficient to attenuate methylation within the promoter of Cgas and more reduction in Cgas promoter methylation was observed when combined with IL-2 stimulation (Fig. 6o). Although IL-2 is well known for driving T cell responses by binding to its

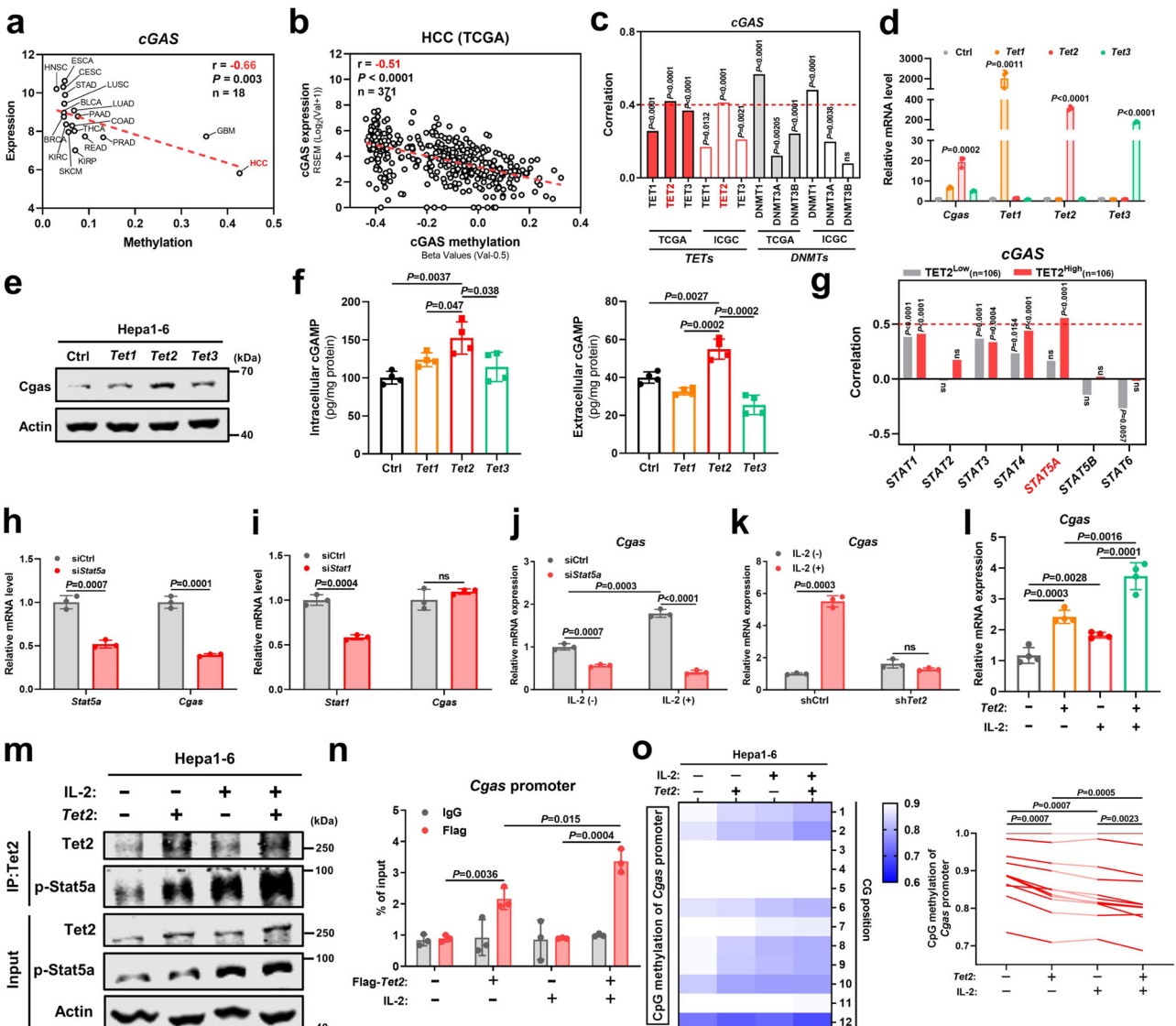

**Fig. 6 | TET2 synergizes with IL-2/STAT5A signaling to promote tumor cGAS expression and cGAMP secretion. a** Correlation between the median expression and methylation of *cGAS* in 18 types of tumors from TCGA datasets. **b** Correlation between *cGAS* expression and methylation in HCC from TCGA dataset ($n = 371$). **c** Correlation between *cGAS* and *TETs* or *DNMTs* expressions in HCC from TCGA ($n = 371$) and ICGC ($n = 212$) datasets. **d** Cgas and *Tet1-3* mRNA levels in Hepa1-6 cells with respective *Tet1-3* overexpression by plasmids ($n = 3$). **e** Cgas protein level in Hepa1-6 cells with respective *Tet1-3* overexpression by plasmids. **f** Intercellular and extracellular cGAMP levels from Hepa1-6 cells with respective *Tet1-3* over-expression by plasmids ($n = 4$). **g** Correlation between *STAT1-6* and *cGAS* expressions in low and high *TET2* expression groups from HCC-ICGC dataset. The low and high expression groups were divided relative to the median expression values. **h** *Stat5a* and *Cgas* mRNA levels in Hepa1-6 cells with *Stat5a* knocked down by siRNA (si*Stat5a*) ($n = 3$). **i** Stat1 and Cgas mRNA levels in Hepa1-6 cells with *Stat1* knocked down by siRNA (si*Stat1*) ($n = 3$). **j** Cgas mRNA level in Hepa1-6-si*Stat5a* or siCtrl cells upon IL-2 stimulation ($n = 3$). **k** Cgas mRNA level in Hepa1-6 cells with *Tet2* knocked down by shRNA (sh*Tet2*) ($n = 3$). **l** Cgas mRNA level in Hepa1-6 cells with *Tet2* overexpression upon IL-2 stimulation (n = 4). **m** Co-IP analysis of Tet2 with p-Stat5a in Hepa1-6 cells with *Tet2* overexpression upon IL-2 stimulation. **n** ChIP-qPCR analysis of Tet2 binding activity to the promoter of *Cgas* in Hepa1-6 cells with Flag-*Tet2* overexpression upon IL-2 stimulation ($n = 3$). **o** Pyrosequencing analysis and quantification of the promoter methylation status of *Cgas* in Hepa1-6 cells with *Tet2* overexpression upon IL-2 stimulation ($n = 3$). *P* values are calculated using two-tailed Pearson correlation coefficient (**a**–**c**, **g**), one-way ANOVA (**d**, **f**, **l**, **n**, **o**) and two-tailed unpaired Student's *t* test (**h**–**k**). ns not significant. Representative of $n = 3$ independent experiments (**e**, **m**). Source data are provided as a Source Data file.

receptors, which consists of three subunits including IL-2Rα (IL2RA), IL-2Rβ (IL2RB), and IL-2Rγ (IL2RG)[29], we found that both IL2RB and IL2RG were expressed in numerous liver cancer cell lines, especially IL2RG, in CCLE database (Supplementary Fig. 8a, b) and similar results were confirmed in liver cancer samples from TCGA dataset (Supplementary Fig. 8c). Since IL2RB and IL2RG are sufficient to form functional receptor for transmitting signals from IL-2[29], liver cancer cells could also respond to IL-2 secreted by T cells and led to STAT5A activation. Together, these results indicate that IL-2-activated STAT5A recruits TET2 binding to the cGAS locus to promote cGAS demethylation and transcription, leading to cGAS upregulation in liver cancer.

## VC-induced TET2 and dsDNA leakage activate tumor cGAS-cGAMP-endothelial STING pathway and promote lymphocyte trafficking

TET2 expression is significantly decreased in HCC, but rarely mutated[30]. Therefore, non-mutational loss of TET2 activity in HCC prompted us to explore the possibility of reactivating TET2 as a strategy to efficiently stimulate cGAS expression. Acting as a cofactor, VC (ascorbic acid or ascorbate) promotes TETs activity and has been proposed as a promising anti-cancer agent[31,32]. Excitingly, VC treatment dramatically induced cGAS expression at both the transcript and protein levels in liver cancer cells in vitro (Fig. 7a, b). Consistent with

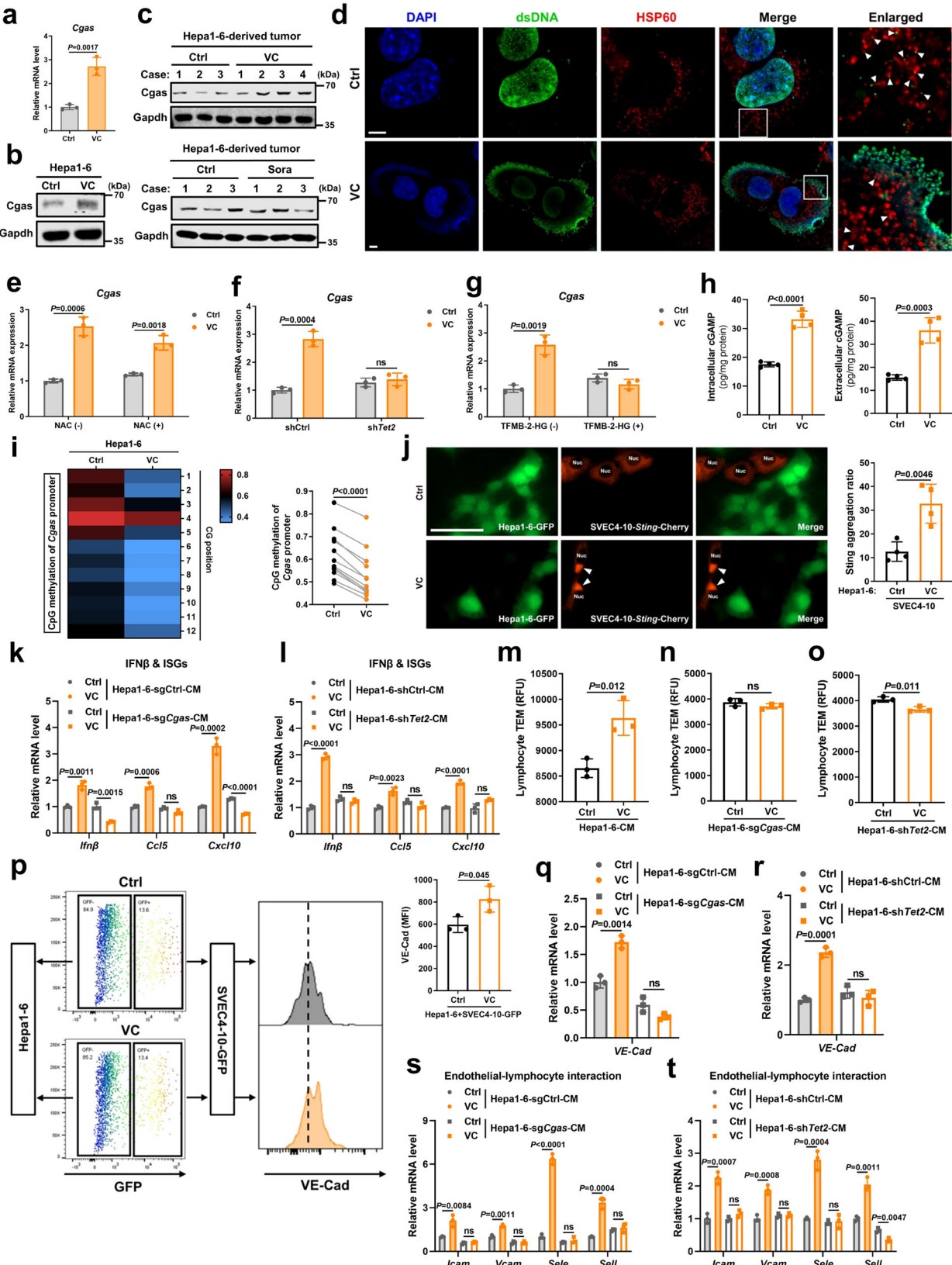

in vitro results, cGAS expression was upregulated in the tumors of mice after VC treatment (Fig. 7c). As a negative control, the multi-target kinase inhibitor sorafenib (Sora), which has been approved as first-line drug for advanced HCC for over a decade[33], had little effect on cGAS expression (Fig. 7c). In addition to Tet enzyme-dependent DNA demethylation, pharmacological VC is known to exert its anti-tumor

activity via hydrogen peroxide-induced oxidative stress[34]. We found that VC treatment induced DNA damage (Supplementary Fig. 9a, b) and cause cytoplasmic leakage of nuclear DNA and mitochondrial DNA (mtDNA) (Fig. 7d), which is known to directly activate cGAS[9,10]. However, inhibition of VC-induced DNA damage, as marker by phosphorylation of histone 2AX (γH2AX), by ROS scavenger N-acetyl-L-cysteine

**Fig. 7 | VC-induced TET2 and dsDNA leakage activate tumor cGAS-cGAMP-endothelial STING pathway and promote lymphocyte trafficking. a, b** *Cgas* mRNA (**a**) and protein (**b**) levels in VC-treated Hepa1-6 cells (*n* = 3). **c** Cgas protein level in Hepa1-6 cells-derived tumor after VC or Sora treatment for 2 weeks. **d** Immunofluorescence costaining of dsDNA, HSP60, and DAPI in VC-treated Hepa1-6 cells. Scale bars, 5 μm. **e** *Cgas* mRNA level in VC-treated Hepa1-6 cells following NAC pretreatment (*n* = 3). **f** *Cgas* mRNA level in VC-treated Hepa1-6 sh*Tet2* cells and shCtrl cells (*n* = 3). **g** *Cgas* mRNA level in VC-treated Hepa1-6 cells following TFMB-2-HG pretreatment (*n* = 3). **h** Intercellular and extracellular cGAMP levels from VC-treated Hepa1-6 cells (*n* = 4). **i** Pyrosequencing analysis and quantification of the promoter methylation status of *Cgas* in VC-treated Hepa1-6 cells (*n* = 3). **j** Immunofluorescence staining and quantification for Sting aggregates in SVEC4-10 cells with *Sting*-Cherry overexpression co-cultured with GFP-labeled Hepa1-6 cells after VC treatment (*n* = 4). The white arrows represent Sting aggregates in SVEC4-10 cells. Scale bars, 50 μm. **k, l** mRNA levels of Ifnβ and ISGs in SVEC4-10 cells after exposure to CM from VC-treated Hepa1-6 sg*Cgas* cells versus sgCtrl cells (**k**) or Hepa1-6 sh*Tet2* cells versus shCtrl cells (**l**) (*n* = 3). **m–o** TEM of mouse splenic lymphocytes through SVEC4-10 cell barrier with pretreatment of CM from Hepa1-6 cells (**m**), Hepa1-6 sg*Cgas* cells (**n**), or Hepa1-6 sh*Tet2* cells (**o**) treated with VC or not (*n* = 3). **p** Flow cytometric analysis and MFI of surface VE-Cad in SVEC4-10-GFP cells co-cultured with Hepa1-6 cells after VC treatment (*n* = 3). **q–t**, mRNA levels of VE-Cad (**q, r**) and indicated genes (**s, t**) in SVEC4-10 cells after exposure to CM from VC-treated Hepa1-6 sg*Cgas* cells versus sgCtrl cells (**q, s**) or Hepa1-6 sh*Tet2* cells versus shCtrl cells (**r, t**) (*n* = 3). *P* values are calculated using two-tailed unpaired Student's *t* test (**a, e–h, j, k–t**) and paired Student's *t* test (**i**). ns, not significant. Representative of *n* = 3 independent experiments (**b–d**). Source data are provided as a Source Data file.

(NAC), failed to reverse the elevated expression of Cgas after VC stimulation (Supplementary Fig. 9a and Fig. 7e). Together, these findings indicate that despite pharmacological VC activates cGAS via ROS-induced dsDNA leakage, VC-mediated cGAS expression is independent of oxidative stress. We further determined whether VC promotes cGAS expression through TET2-dependent DNA demethylation. Tet2 blockade by either Tet2 knockdown or Tet enzyme inhibitor (cell-permeable 2-hydroxyglutarate (2-HG)) significantly decreased VC-induced Cgas expression in liver cancer cells (Fig. 7f, g). Consistent with Tet2 overexpression (Fig. 6o), VC treatment obviously reduced the methylation level of Cgas promoter (Fig. 7i). Collectively, these findings imply that pharmacological VC not only activates tumor cGAS through cytoplasmic dsDNA leakage, but also increases tumor cGAS expression via epigenetic modifications by Tet2-dependent DNA demethylation.

To next assessed whether VC-induced tumoral cGAS activates endothelial STING and enhances lymphocyte trafficking, an obvious increase in cGAMP production and secretion in liver cancer cells after VC treatment was firstly confirmed (Fig. 7h). VC treatment resulted in much more marked formation of perinuclear Sting aggregates, which are readout for STING activation by cGAMP[10,18], in *Sting*-Cherry-expressed endothelial cells when co-cultured with GFP-labeled Hepa1-6 cells (Fig. 7j), whereas VC treatment on endothelial cells alone neither induced more Sting aggregates nor upregulated Cgas expression (Supplementary Fig. 10a–c), implying that VC indirectly activates endothelial Sting through cancer cells. Similarly, CM derived from VC-treated Hepa1-6 cells significantly activated endothelial Sting, as evidenced by elevated phosphorylation of Sting and Tbk1, as well as increased Ifnβ and ISGs expressions (Supplementary Fig. 9c, d), whereas VC treatment on endothelial cells alone led to no obvious difference in endothelial Sting activation (Supplementary Fig. 10d, e). Importantly, when the Cgas or Tet2 expression was blocked (Supplementary Fig. 9e), these cells-derived CM after VC stimulation failed to activate endothelial STING (Supplementary Fig. 9f, g and Fig. 7k, l). Furthermore, we identified that more lymphocytes transmigrated through endothelial cell barriers stimulated by CM from VC-treated Hepa1-6 cells versus PBS-treated cells (Fig. 7m). After exposure to CM from VC-treated Hepa1-6 cells with Cgas or Tet2 deficiency, endothelial cell barriers did not promote transendothelial migration of lymphocytes (Fig. 7n, o). Therefore, these results suggest that VC-induced TET2 upregulates tumor cGAS to activate STING pathway in endothelial cells and promote lymphocyte trafficking.

Mechanistically, after exposure to VC, endothelial cells in mixed culture with Hepa1-6 cells displayed increased VE-Cad expression, a marker of endothelial cell junction stability, whereas VC treatment on endothelial cells alone had no notable effect on their VE-Cad expression (Supplementary Fig. 10f). Similarly, CM derived from VC-treated Hepa1-6 cells upregulated VE-Cad and endothelial-lymphocyte interaction-associated adhesion molecules in endothelial cells compared with CM derived from PBS-treated Hepa1-6 cells (Fig. 7p and

Supplementary Fig. 9h, i), whereas no change in endothelial-lymphocyte interaction-associated genes were observed in endothelial cells after direct VC treatment (Supplementary Fig. 10g). In contrast, disruption of either Cgas or Tet2 abrogated increased VE-Cad and endothelial-lymphocyte interaction-associated adhesion molecules expressions in endothelial cells induced by CM derived from VC-treated Hepa1-6 cells (Fig. 7q–t). Together, these data indicate that VC induces TET2 activation to upregulates tumor cGAS, further producing cGAMP to activate endothelial STING and enhance transendothelial migration of lymphocytes via maintenance of endothelial cell junction stability and upregulation of endothelial-lymphocyte interaction-associated adhesion molecules.

## Tumor TET2-p-STAT5A-cGAS-LRRC8C-endothelial STING axis correlates with vascular normalization and immune infiltration in human liver cancer

Next, we investigated the protein expression levels of tumor TET2-p-STAT5A-cGAS-LRRC8C-endothelial STING axis in tumor biopsies from liver cancer patients. In serial sections of HCC samples from the same patient, high tumor LRRC8C expression was associated with high STING expression in CD31⁺ vascular endothelial cells in either high or low tumor cGAS expression group (Fig. 8a, b). Importantly, when tumor LRRC8C expression was high, endothelial STING expression was significantly higher in high tumor cGAS expression group versus low tumor cGAS expression group (Fig. 8a, b), indicating that tumor cGAS expression correlates with endothelial STING activation depending on LRRC8C.

Compared with tumors with either low cGAS or LRRC8C expression, α-SMA⁺ pericyte coverage of CD31⁺ tumor vessels, which is one of the hallmarks of tumor vessel normalization[15], was dramatically increased in tumors with both high cGAS and LRRC8C expressions (Fig. 8a, c). Tumor LRRC8C expression negatively correlated with vascular invasion when cGAS was highly expressed, whereas no obvious correlation was shown in low cGAS expression group (Fig. 8a, d). Consistently, cGAS and LRRC8C expressions were downregulated in eight of 11 portal vein tumor thrombosis (PVTT) in comparison to corresponding primary tumors (Fig. 8e). Furthermore, high tumor LRRC8C expression was associated with low hypoxia marker GLUT1 expression in the high cGAS expression group, but not when cGAS was low expressed in HCC tissues (Fig. 8f, g). To further assess immune infiltration, tumor LRRC8C expression positively correlated with intratumoral CD8⁺ T cell infiltration, especially when tumor cGAS expression was high (Fig. 8f, h). Together, these results suggest that tumor cGAS-LRRC8C-endothelial STING axis was associated with vascular normalization and immune infiltration in human liver cancer.

We further verified the regulation of tumor cGAS by TET2 and STAT5A signaling in human liver cancer samples and found that high tumor TET2 and p-STAT5A expressions were associated with high tumor cGAS expression (Fig. 8i, j). In TCGA database, tumor TET2, STAT5A, and cGAS expressions positively correlated with vascular

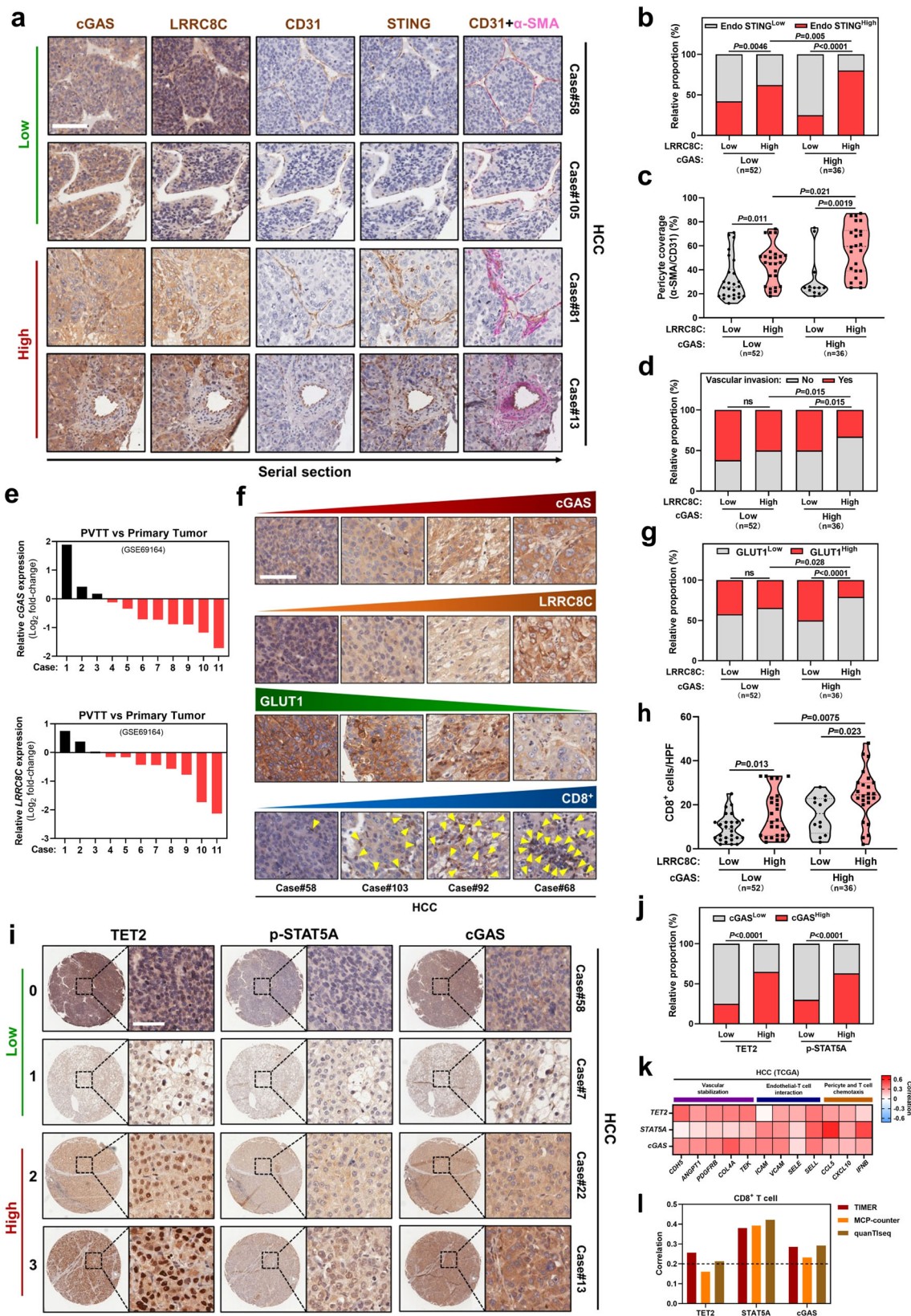

normalization-associated genes including vascular stabilization, endothelial-lymphocyte interaction, pericytes and T cell chemotaxis (Fig. 8k). Additionally, a positive correlation was found between tumor TET2, STAT5A, cGAS expression and intratumoral CD8+ T cell infiltration (Fig. 8l). Furthermore, HCC patients with high tumor TET2,

p-STAT5A, cGAS, LRRC8C, and endothelial STING expression had better clinical outcome (Supplementary Fig. 11a–e). Collectively, these data show that tumor TET2-p-STAT5A-cGAS-LRRC8C-endothelial STING axis is linked to vascular normalization and immune infiltration in human liver cancer.

**Fig. 8 | Tumor TET2-p-STAT5A-cGAS-LRRC8C-endothelial STING axis correlates with vascular normalization and immune infiltration in human liver cancer.** Representative immunohistochemical images (**a**) and correlation analysis of tumor cGAS and LRRC8C expressions with endothelial STING expression (**b**), pericyte (α-SMA⁺) coverage of vessels (CD31⁺) (**c**), and vascular invasion (**d**) in serial sections of HCC samples from the same patient (*n* = 88). Based on the intensity of staining (protein expression), the patients were subdivided into two groups: low (staining score 0-1) and high (staining score 2-3) expression group. Scale bars, 100 μm. **e** *cGAS* and *LRRC8C* mRNA levels between PVTT and the corresponding primary HCC samples (*n* = 11 pairs) (GEO: GSE69164). Representative immunohistochemical images (**f**) and correlation analysis of tumor cGAS and LRRC8C expressions with GLUT1 expression (**g**) or CD8⁺ T cell infiltration (**h**) in human liver cancer samples

(*n* = 88). The dashed line across the violin plots represents the quartiles and the full line depicts the median. Scale bars, 100 μm. Representative immunohistochemical images (**i**) and correlation analysis (**j**) of tumor TET2 and p-STAT5A expressions with cGAS expression in human liver cancer samples (*n* = 88). Scale bars, 50 μm. **k** Pearson's correlation of *TET2*, *STAT5A*, and *cGAS* expressions with genes related to vascular stabilization, endothelial-T cell interaction, and pericyte/T cell chemotaxis in HCC from TCGA dataset (*n* = 371). **l** Pearson's correlation of *TET2*, *STAT5A*, and *cGAS* expressions with CD8⁺ T cell infiltration in HCC from TCGA dataset (*n* = 371) through TIMER, MCP-counter, and quanTIseq. *P* values are calculated using Chi-square test (**b**, **d**, **g**, **j**) and one-way ANOVA (**c**, **h**). ns, not significant. Source data are provided as a Source Data file.

## VC treatment induces vascular normalization and boosts efficacy of anti-PD-L1 therapy or anti-PD-L1 combination therapy with IL-2

We next investigated the impact of high-dose VC treatment on tumor vasculatures and immune infiltration in vivo. Intraperitoneal injection of 4 g/kg VC (equivalent to ~1.3 g/kg intravenously), a pharmacological dose widely used in mouse models and showing inhibitory effects against various tumors growth including liver cancer[34,35], was administered to immunocompetent mice subcutaneously bearing Hepa1-6 tumors (Supplementary Fig. 12a, b). Similar to the effect caused by tumor cGAS and host STING activation, VC treatment led to a reduction in CD31⁺ tumor vessels and intratumoral hypoxia, as evidenced by decreased GLUT1 expression, but an obvious increase in α-SMA⁺ pericyte coverage of CD31⁺ tumor vessels and intratumoral CD8⁺ T cells infiltration (Fig. 9a, Supplementary Fig. 12e). In accordance with in vitro findings, downstream of STING activation IFNβ and ISGs, vascular stabilizing genes, endothelial-lymphocyte interaction-associated adhesion molecules were up-regulated in VC-treated tumors compared with PBS-treated tumors (Supplementary Fig. 12f). In contrast, although HCC first-line drug Sora treatment also delayed tumor growth and reduced tumor vessel density in mice, it failed to normalize tumor vessels, as shown by no difference in pericyte coverage of tumor vessels and enhanced intratumoral hypoxia, and thereby led to decreased CD8⁺ T cells infiltration (Fig. 9a, Supplementary Fig. 12c–e). This is in line with previous observations that Sora promoted hypoxia-mediated immunosuppression[36,37]. Therefore, these results suggest that VC treatment, but not Sora, induces normalization of the tumor vasculature and improves immune infiltration in liver cancer.

We evaluated a recent HCC dataset with patients who underwent anti-PD-1/PD-L1 therapy[38] and found that 33–50% patients with high ISGs expressions benefitted from anti-PD-1/PD-L1 treatment, whereas only 8–16% patients with low ISGs expressions responded to anti-PD-1/PD-L1 therapy (Supplementary Fig. 13a), imply that cGAS-STING activation may improve the efficacy of immune checkpoint blockade in liver cancer. To exclude that these observations resulted from an impact of tumor size on efficacy of anti-PD-L1 therapy, we adjusted cell numbers of Cgas overexpression cells and chose tumors with similar sizes to Ctrl tumors for anti-PD-L1 therapy. As expected, compared with Ctrl group, anti-PD-L1 treatment resulted in much smaller tumors (Supplementary Fig. 13b, c), coincided with more intratumoral CD8⁺ cytotoxic T cells infiltration in Cgas overexpression group with similar sizes to Ctrl tumors before therapy (Supplementary Fig. 13d). Therefore, these findings confirm the enhanced efficacy of anti-PD-L1 therapy by cGAS expression in liver cancer regardless of tumor size.

Since VC is a stimulator of tumor cGAS and endothelial STING activation, as shown in Fig. 6, we tested whether VC treatment sensitizes liver cancer to immunotherapy. In subcutaneously implanted tumor models, striking tumor regressions were observed in mice treated with the combination of VC and anti-PD-L1 in comparison to the single VC or anti-PD-L1 treatment (Fig. 9b, c), suggesting that VC augments PD-L1 checkpoint inhibitor-induced anti-tumor responses. In contrast, consistent with its role in immunosuppression, Sora

treatment displayed no additional anti-tumor activity when combined with anti-PD-L1 antibody (Fig. 9d, e), similar to previous findings[36]. Of note, the combination therapy of VC and anti-PD-L1 exhibited a notable achievement of complete tumor repression approximately 28 days following a treatment period of 21 days in mice (Supplementary Fig. 15a). Therefore, these results indicate that VC combined with anti-PD-L1 treatment achieves complete tumor repression after a long period of therapy. Because subcutaneously implanted tumor models may not fully represent the pathologic microenvironment, we further employed orthotopic liver cancer models, a more reliable representative of human liver cancer[39]. Less infiltration of immune cells including CD8⁺ T cells, NK cells, and macrophages were found in subcutaneous versus orthotopic models (Supplementary Fig. 14a–d), indicating that the therapeutic efficacy of anti-PD-L1 may be diminished in orthotopic models. Nevertheless, in line with the results in subcutaneous xenograft models, combined treatment of VC and anti-PD-L1 led to significant smaller tumor burdens than the single IgG Ctrl, VC or anti-PD-L1-treated group in orthotopic tumor models (Fig. 9f–h).

Based on the synergy between TET2 to IL-2/STAT5A signaling (Fig. 6), we further examined whether VC enhanced the efficacy of anti-PD-L1 and IL-2 combination therapy, which is a potentially promising candidate for cancer therapy[40,41]. Excitingly, in addition to the combined anti-tumor effect of VC and anti-PD-L1, VC treatment further boosted efficacy of anti-PD-L1 combination therapy with IL-2 in both subcutaneously implanted tumor models (Fig. 9i–k) and orthotopic liver cancer models (Fig. 9l–n). Remarkably, we observed that this enhanced effect was comparable between orthotopic and subcutaneous tumor models. Collectively, VC treatment induces vascular normalization and boosts efficacy of anti-PD-L1 therapy or anti-PD-L1 combination therapy with IL-2.

## Tumor TET2-STAT5A-cGAS-host STING axis mediates VC-induced vascular normalization and therapeutic efficacy of VC combined with anti-PD-L1

To assess whether this synergistic anti-tumor effect is mediated through CD8⁺ T cell population, tumor-bearing mice were treated with VC and anti-PD-L1 antibody in the presence of either IgG Ctrl or anti-CD8 antibody to deplete CD8⁺ T cells (Supplementary Fig. 15b). As expected, the enhanced tumor regressions by combined therapy of VC and anti-PD-L1 coincided with increased infiltration of intratumoral CD8⁺ cytotoxic T cells in IgG Ctrl groups (Supplementary Fig. 15c–e). However, after CD8 depletion, confirmed by flow cytometry (Supplementary Fig. 15f), the improved anti-tumor benefit of VC combined with anti-PD-L1 was absent in comparison to single VC or anti-PD-L1 treatment (Supplementary Fig. 15c–e), suggesting the combinational efficiency of VC and anti-PD-L1 therapy depending on CD8⁺ T cell-induced anti-tumor immune response.

Importantly, in Tet2 or Stat5a deficiency cells-derived tumors, the combination of VC and anti-PD-L1 had no obvious changes in tumor burdens as compared to single VC or anti-PD-L1 therapy (Fig. 10a–d). Moreover, VC therapy did not exert a significant impact on tumor vessel coverage by pericytes, intratumoral hypoxia levels, and CD8⁺ T

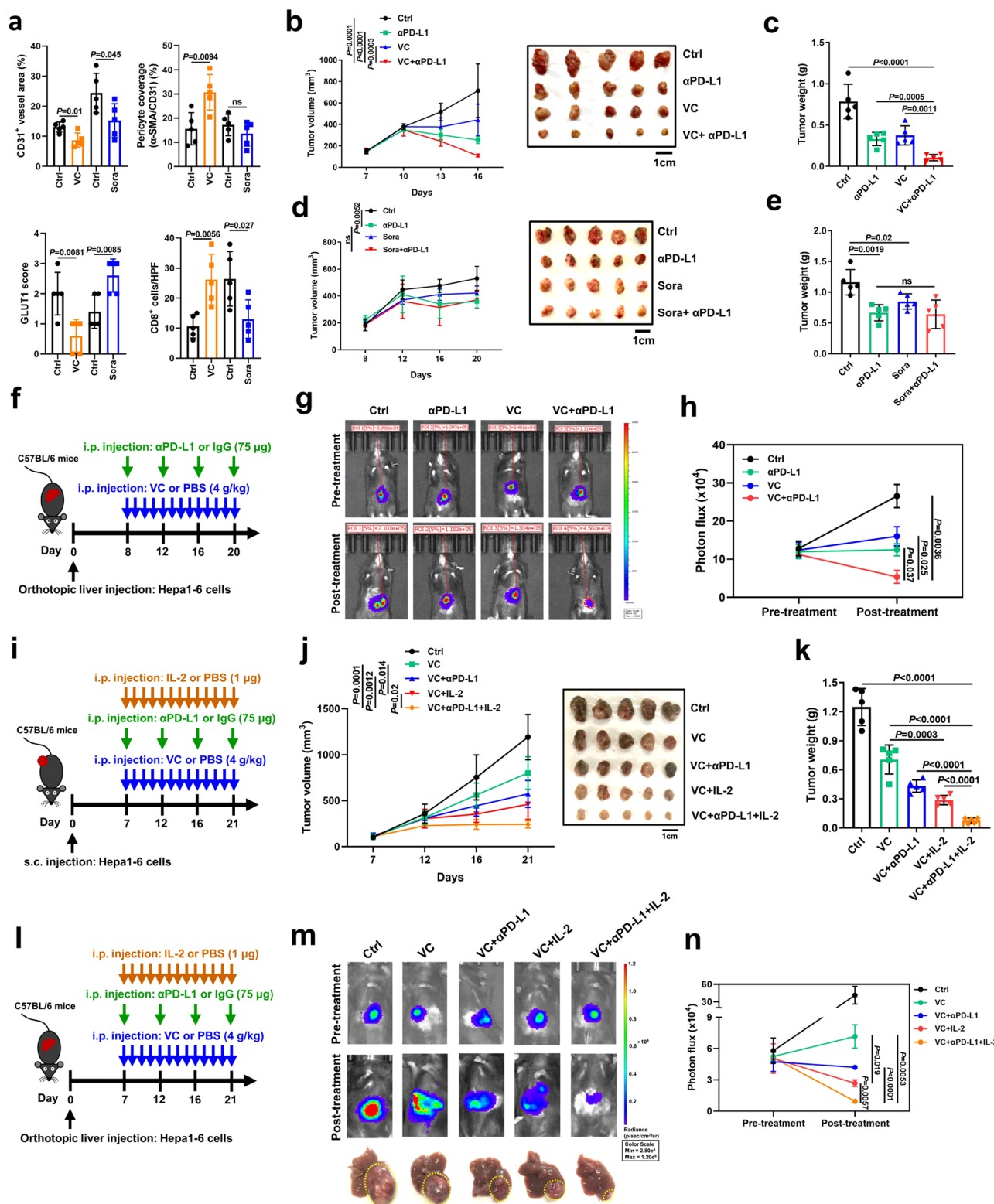

cell infiltration in these tumors with Tet2 or Stat5a deficiency (Fig. 10e, Supplementary Fig. 12g). Furthermore, the knockdown of Tet2 or Stat5a in tumors negated the upregulation induced by VC treatment of IFNβ and ISGs, vascular stabilizing genes, endothelial-lymphocyte interaction-associated adhesion molecules (Supplementary Fig. 12h). Together, these findings provide conclusive evidence for the crucial involvement of tumor Tet2 and Stat5a in VC-mediated vascular normalizing effects and the combinational efficiency of VC and anti-PD-L1 therapy in liver cancer.

Furthermore, we depleted tumor cGAS and employed STING inhibitor C-176 to block host STING activation. Given the initial low expression level of Cgas in native Hepa1-6 cells (Supplementary Fig. 1a), Cgas knockout (KO) in Hepa1-6 cells exhibited a marginal augmentation of tumor growth and burdens, which did not reach statistical significance (Supplementary Fig. 15g, h). Nevertheless, Cgas KO in Hepa1-6 cells completely abrogated Cgas upregulation induced by VC treatment and in tumors derived from cGAS KO cells, the combination of VC and anti-PD-L1 had no obvious changes in tumor

**Fig. 9 | VC treatment induces vascular normalization and boosts efficacy of anti-PD-L1 therapy or anti-PD-L1 combination therapy with IL-2.**
**a** Quantification for CD31$^+$ vessels density, pericyte (α-SMA$^+$) coverage of vessels (CD31$^+$), GLUT1$^+$ hypoxic area, and CD8$^+$ T cells in VC-treated or Sora-treated tumors ($n = 5$). Tumor growth curves (**b**) and tumor burdens (**c**) of C57BL/6 mice injected subcutaneously with Hepa1-6 cells with treatment of αPD-L1 and VC either alone or in combination ($n = 5$ mice per group). Tumor growth curves (**d**) and tumor burdens (**e**) of C57BL/6 mice injected subcutaneously with Hepa1-6 cells with treatment of αPD-L1 and Sora either alone or in combination ($n = 5$ mice per group). Scheme representing the experimental procedure (**f**), representative luciferase-based bioluminescence images (**g**) and quantification (**h**) of C57BL/6 mice injected in situ with Hepa1-6 cells with pre- and post-treatment of αPD-L1 and VC either alone or in combination ($n = 3$ mice per group). Scheme representing the experimental procedure (**i**), tumor growth curves (**j**), and tumor burdens (**k**) of C57BL/6 mice injected subcutaneously with Hepa1-6 cells with treatment of VC alone or combined with αPD-L1, IL-2, or αPD-L1 + IL-2 ($n = 5$ mice per group). Scheme representing the experimental procedure (**l**), representative luciferase-based bioluminescence images (**m**) and quantification (**n**) of C57BL/6 mice injected in situ with Hepa1-6 cells with pre- and post-treatment of VC alone or combined with αPD-L1, IL-2, or αPD-L1 + IL-2 ($n = 3$ mice per group). *P* values are calculated using two-tailed unpaired Student's *t* test (**a**), two-way ANOVA (**b**, d, **j**) and one-way ANOVA (**c**, **e**, **h**, **k**, **n**). ns not significant. Source data are provided as a Source Data file.

burdens as compared to single VC or anti-PD-L1 therapy (Fig. 10f–h). Similarly, host STING inhibition by C-176 treatment also abrogated the synergistic anti-tumor effect of VC and PD-L1 blockade in vivo (Fig. 10i–k). Moreover, VC therapy had no substantial effect on tumor vessel coverage by pericytes, intratumoral hypoxia, and CD8$^+$ T cell infiltration after tumor cGAS deficiency and host STING inhibition (Fig. 10l, Supplementary Fig. 12i). Additionally, tumoral cGAS KO or host STING inhibition nullified the VC-induced upregulation of IFNβ and ISGs, vascular stabilizing genes, endothelial-lymphocyte interaction-associated adhesion molecules in tumors (Supplementary Fig. 12j). Together, these results confirm the vital role of tumor cGAS and host STING activation in VC-mediated vascular normalizing effects and the combinational efficiency of VC and anti-PD-L1 therapy in liver cancer. Overall, our findings demonstrate that tumor TET2-STAT5A-cGAS-host STING axis mediates VC-induced vascular normalization and therapeutic efficacy of VC combined with anti-PD-L1.

## Discussion

cGAS-STING pathway is vital for anti-tumor immunity[11,12]. In addition to the well-studied anti-tumor immune effect of the cGAS-STING pathway in immune cells[13], cancer cell-intrinsic cGAS-STING pathway activation has also been recently shown to define their immunogenicity and make tumors hot[42,43]. However, the crosstalk of cGAS-STING pathway between tumor cells and host cells in tumor microenvironment remains largely unclear. On one side, this work demonstrates that cancer cell-intrinsic expression of cGAS, but not STING, determines tumor vascular normalization and anti-tumor immune response in a host STING-dependent manner. On the other side, host cGAS is dispensable but tumor cGAS is necessary for host STING-mediated vascular normalization and anti-tumor immunity. Together, we identify the interdependence between tumor cGAS and host STING in the regulation of vasculature remodeling and anti-tumor immunity in liver cancer. Mechanistically, tumor cGAS produces cGAMP, which further transports via LRRC8C channels to activate STING in endothelial cells, enhancing recruitment and transendothelial migration of lymphocytes. In turn, intratumoral infiltrating lymphocytes secrete IL-2 to activate tumor STAT5A signaling, which further synergizes with TET2 to epigenetically upregulate tumor cGAS, indicating a positive feedback loop. Accordingly, stimulating tumor TET2 by VC to accelerate this positive feedback loop may effectively induce tumor vascular normalization and boost immunotherapy efficacy.

Currently, intratumoral STING activation with STING agonists has been reported to normalize tumor vasculature[44,45]. However, whether tumor STING or host STING plays a dominant role remains to be defined. Using Sting-deficient mice, we identify that host STING, rather than cancer cell-intrinsic STING, determines vascular normalization and anti-tumor immune response in highly angiogenic liver cancer. This phenomenon may be explained by several reasons. First, in addition to the recruitment and activation of T cells and pericytes by STING pathway target genes IFNβ and ISGs such as CCL5 and CXCL10, endothelial STING pathway plays a crucial role in regulating T cells trafficking and endothelial cell function. For instance, the activated STING downstream transcriptional factor IRF3 has been shown to bind

to the promoter of ICAM-1 and induce ICAM-1 expression, one of the key adhesion molecules for transendothelial migration of T cells, in endothelial cells[46]. Similarly, we find that blocking the IFNβ signaling pathway by STAT1 activation inhibitor fails to reverse cGAMP-induced ICAM-1 expression and IFNβ treatment has little effect on ICAM-1 expression in endothelial cells, suggesting the regulation of endothelial ICAM-1 expression mainly by STING pathway, rather than its target genes IFNβ and ISGs. Additionally, recent studies have revealed that when endothelial STING pathway is activated, the downstream kinase TBK1 inhibits endothelial cell proliferation via suppressing YAP pathway[47]. Therefore, although STING activation in cancer cells may also enhance IFNβ and ISGs production for T cell recruitment, it does not alter the expression of adhesion molecules in endothelial cells or the transendothelial migration of T cells. Furthermore, since endothelial cells have been demonstrated to be the principal source of type I IFNs in growing tumors and endothelial cell-derived IFNβ initiates CD8$^+$ T cell-mediated anti-tumor immunity[48,49], the contribution of IFNβ produced by cancer cells to T cell priming may be minimal. Collectively, endothelial STING is superior to tumor STING in anti-tumor immunity, as STING activation in endothelial cells not only enhances T cells recruitment and activation, but also promotes tumor vascular normalization for T cells trafficking.

STING is classically activated by cGAS-produced cGAMP and cGAMP is known to be transmitted between cells[15,16]. Therefore, host STING activation and its vascular normalizing effect may be controlled by tumor cGAS or host cGAS. Surprisingly, we find that host STING-mediated vascular normalization and anti-tumor immune response depends on tumor cGAS, but not host cGAS. It is well accepted that cancer cells are not only the major component of tumor, but also are often constitutively rich in cytoplasmic dsDNA, which further increases upon DNA damaging therapies such as chemotherapy to directly activate cGAS to produce cGAMP[50]. Therefore, cancer cells are the dominant source of cGAMP in the tumor microenvironment, especially when their cGAS expressions are upregulated by TET2 in this study. Furthermore, we show that most liver cancer cells express cGAS, but hardly express STING, in HCC samples, consistent with previous reports showing loss of cancer cell-intrinsic STING in melanoma and colorectal cancer[51,52]. In contrast, host cells, especially endothelial cells, in tumor microenvironment exhibit distinct STING expression. Since cGAMP is known to be transmitted between cells[15,16], cGAMP produced by cGAS in cancer cells, which are frequently lack of intrinsic STING, is prone to be secreted to tumor microenvironment, leading to host STING activation. These reasons may explain why tumor cGAS is essential for host STING activation and thereby host STING-mediated vascular normalization and anti-tumor immune response, whereas cGAS of host cells is dispensable.

cGAS functions as an essential DNA sensor, which senses the cytoplasmic dsDNA and activates the anti-tumor immune response[9,11]. However, the regulatory mechanism of cGAS expression and activity remained to be fully understood. Hypermethylation of the cGAS gene promoter is associated with its downregulation in liver cancer[26], but the methylation-dependent regulatory mechanism is unclear. The present work identifies TET2 methylcytosine

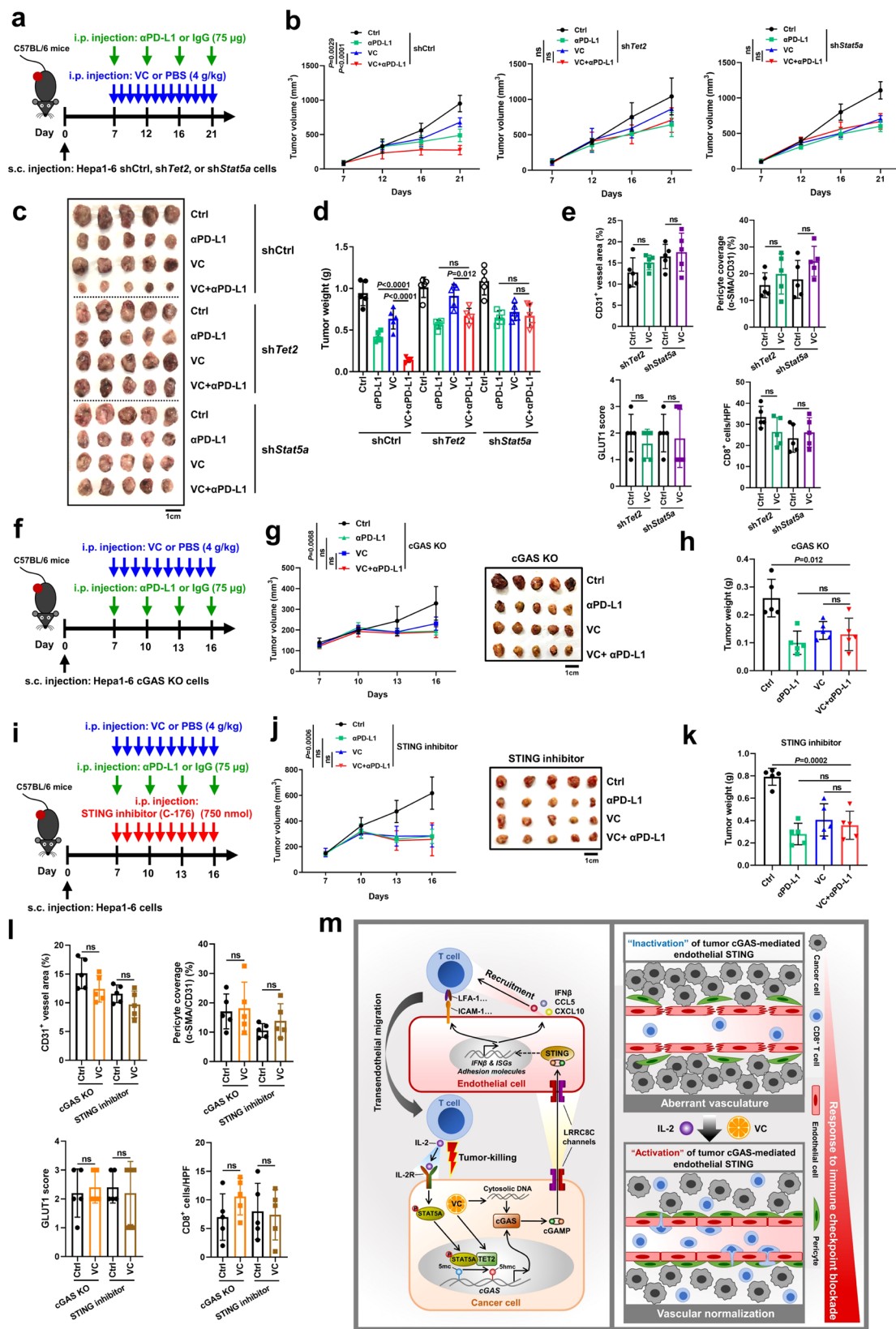

dioxygenase as a responsible enzyme for cGAS demethylation and reveal that activated STAT5A recruits TET2 binding to the cGAS locus to promote cGAS demethylation and transcription, resulting in cGAS upregulation in liver cancer cells. Expectedly, we further demonstrate that as a TETs enzyme activator, VC treatment increases cGAS expression via TET2-dependent demethylation in liver cancer cells.

TET2-mediated cGAS expression may be cancer cell-specific, as VC exposure has no obvious effect on cGAS expression in endothelial cells. In addition to TET2-mediated demethylation, further investigation is needed to clarify whether the DNMT family plays a predominant role in influencing the DNA methylation status of the cGAS promoter in our model.

**Fig. 10 | Tumor TET2-STAT5A-cGAS-host STING axis mediates VC-induced vascular normalization and therapeutic efficacy of VC combined with anti-PD-L1.** Scheme representing the experimental procedure (**a**), tumor growth curves (**b**), tumor images (**c**), and tumor burdens (**d**) of C57BL/6 mice injected subcutaneously with Hepa1-6-sh*Tet2* cells, sh*Stat5a* cells, or shCtrl cells with treatment of αPD-L1 and VC either alone or in combination (*n* = 5). **e** Quantification for CD31⁺ vessels density, pericyte (α-SMA⁺) coverage of vessels (CD31⁺), GLUT1⁺ hypoxic area, and CD8⁺ T cells infiltration in sh*Tet2* and sh*Stat5a* tumors after VC treatment (*n* = 5). Scheme representing the experimental procedure (**f**), tumor growth curves (**g**), and tumor burdens (**h**) of C57BL/6 mice injected subcutaneously with Hepa1-6-sg*Cgas* cells (cGAS KO) with treatment of αPD-L1 and VC either alone or in combination (*n* = 5). Scheme representing the experimental procedure (**i**), tumor growth curves (**j**), and tumor burdens (**k**) of C57BL/6 mice injected subcutaneously with Hepa1-6 cells with treatment of αPD-L1 and VC either alone or in combination after STING inhibitor C-176 treatment (*n* = 5). **l** Quantification for CD31⁺ vessels density, pericyte

(α-SMA⁺) coverage of vessels (CD31⁺), GLUT1⁺ hypoxic area, and CD8⁺ T cells infiltration in cGAS KO and STING inhibitor-treated tumors after VC treatment (*n* = 5). **m** Schematic illustration of TET2-mediated tumor cGAS triggering endothelial STING activation to regulate vasculature remodeling and anti-tumor immunity. Left: tumor cGAS produces cGAMP, which further transports via LRRC8C channels to activate STING in endothelial cells, enhancing recruitment and transendothelial migration of lymphocytes. In turn, intratumoral infiltrating lymphocytes secrete IL-2 to activate tumor STAT5A signaling, which further synergizes with TET2 to epigenetically upregulate tumor cGAS, indicating a positive feedback loop. Right: stimulating tumor TET2 by VC to accelerate this positive feedback loop may effectively induce tumor vascular normalization and boost immunotherapy efficacy. *P* values are calculated using two-way ANOVA (**b**, **g**, **j**), one-way ANOVA (**d**, **h**, **k**), and two-tailed unpaired Student's *t* test (**e**, **i**). ns, not significant. Source data are provided as a Source Data file.

Emerging evidences show that TET methylcytosine dioxygenases regulate tumor immunity[53,54]. We have demonstrated that NAD⁺ metabolism drives IFNγ-induced PD-L1 checkpoint expression via TET1 and promotes tumor immune evasion[53]. In addition to PD-L1 induction, TET2 has been shown to upregulate chemokines and increase tumor-infiltrating lymphocytes[54]. Collectively, tumoral TETs have been shown to directly regulate interaction between tumor cells and immune cells via immune checkpoint and chemokines so far. However, the highly abnormal and dysfunctional vasculature of tumors is a crucial barrier of immune cell infiltration into tumor parenchyma, thus limiting anti-tumor immune responses[55]. Therefore, the role of TETs in tumor vasculature remains to be dissected. The present study shows that TET2 upregulates tumor cGAS expression to produce cGAMP, which further activates STING pathway in endothelial cells, leading to tumor vascular normalization and enhanced intratumoral CD8⁺ T cell infiltration. Our findings uncover the role of tumor suppressor TET2 in anti-tumor immunity via cGAS-STING-mediated vascular normalization.

Pharmacologic VC has recently emerged as a promising anti-cancer agent with low toxicity and financial cost according to different pre-clinical and clinical trials[34,56]. VC is known to exert anti-tumor activity through hydrogen peroxide-induced oxidative stress to preferentially kill cancer cells when used at high-doses or through control of TET enzymes activity to restore the abnormal DNA methylation pattern observed in many tumors[31,32]. To date, a series of phase I/II clinical trials have demonstrated the safety and efficacy of intravenous high-dose VC as a monotherapy or in combination with radiation and conventional chemotherapies for various cancers[55]. We have previously shown that pharmacologic VC preferentially eradicates liver cancer stem cells and intravenous VC use is linked to improved survival of HCC patients[35]. Although high-dose VC has currently been reported to enhance immunotherapy[57,58], yet the immunomodulatory effects of pharmacologic VC have been incompletely studied. In this work, we provide evidence that stimulating TET2 activity by VC combined with VC-induced dsDNA leakage activate tumor cGAS-cGAMP-endothelial STING pathway and promote lymphocyte trafficking. There are various mechanisms that trigger the export of damaged DNA fragments derived from the nucleus and/or mitochondria, such as formation of abnormal micronucleus, mitochondrial degeneration and membrane potential reduction, autophagy[59,60]. Further studies are needed to clarify the mechanisms underlying how VC treatment triggers the export of damaged DNA fragments derived from the nucleus and/or mitochondria, and to ascertain the specificity of this modulation towards tumor cells. Importantly, we further reveal that VC treatment induces tumor vascular normalization and boosts efficacy of anti-PD-L1 therapy or anti-PD-L1 combination therapy with IL-2 in vivo, provides promising strategies to enhance the combinational immunotherapy for liver cancer.

In summary, our findings elucidate a crosstalk between tumor and vascular endothelial cells in the tumor microenvironment via cGAS and

STING essential for tumor vascular normalization and anti-tumor immunity (Fig. 10m). Based on the epigenetic regulation of tumor cGAS by TET2 synergized with STAT5A signaling, stimulating TET2 activity by VC triggers tumor cGAS-cGAMP-endothelial STING pathway activation and induces tumor vascular normalization, thereby potentiating immunotherapy efficacy. This work not only uncovers the epigenetic regulatory mechanism of tumor cGAS, but also reveals the immunomodulatory roles of VC and TET2 in liver cancer, providing a strategy to induce tumor vascular normalization and a scientific rationale for further clinical trials combining VC with immunotherapy.

## Methods
### Study approval
All animal protocols used in this study were approved by The University Committee on Use and Care of Animals of Second Military Medical University. Tumor tissues from patients with HCC were obtained from the Eastern Hepatobiliary Surgery Hospital (EHBH), Shanghai, China, with male to female ratio of 6:1, conforming to the sex disparities in HCC, which has a strong male predominance[1]. Patient consent was obtained prior to the commencement of the study. All procedures performed in the study were approved by the Ethical Committee of the Second Military Medical University and in accordance with the Declaration of Helsinki.

### Cell culture
The murine liver cancer cell line Hepa1-6 (SCSP-512) and human HCC cell lines Huh7 (SCSP-526) were purchased from Cell Bank of Type Culture Collection of Chinese Academy of Sciences (CBTCCCAS, Shanghai, China). The murine endothelial cell line SVEC4-10 (CRL-2181) was purchased from American Type Culture Collection (ATCC; Manassas, VA, USA). Cell lines were authenticated by STR profiling and verified to be mycoplasma negative. All cells were grown in DMEM (Gibco) with 10% FBS at 37 °C and 5 % CO₂.

### Antibodies and reagents
Anti-human cGAS (79978, 1:1000), anti-mouse cGAS (31659, 1:1000), anti-human/mouse STING (13647, 1:1000), anti-mouse p-STING (Ser365) (72971, 1:1000), anti-human/mouse TBK1 (38066, 1:1000), anti-human/mouse p-TBK1(Ser172) (5483, 1:1000), anti-human/mouse α-SMA (19245, 1:100) antibodies, and anti-human/mouse γH2AX (9718, 1:1000) were purchased from Cell Signaling Technology. Anti-human/mouse LRRC8C (21601-1-AP, 1:1000), anti-human/mouse Flag (20543-1-AP, 1:1000), anti-human/mouse GAPDH (60004-1-Ig, 1:10,000), and anti-human/mouse β-Actin (66009-1-Ig, 1:10000) were purchased from Proteintech. Anti-human/mouse TET2 (GTX124205, 1:1000) and anti-human/mouse p-STAT5A (GTX13593, 1:1000) were purchased from GeneTex. Anti-human/mouse CD31 (ab182981, 1:100), anti-human/mouse GLUT1 (ab115730, 1:500), anti-human CD8 (ab93278, 1:100), anti-mouse CD8 (ab209775, 1:100), anti-dsDNA (ab27156,

1:500), anti-mouse NKp46 (ab233558, 1:200), and anti-human/mouse HSP60 (ab46798, 1:200) were purchased from Abcam. Anti-human/mouse VEGFR2 (sc-6251, 1:1000) was purchased from Santa Cruz. Anti-mouse VE-Cad (CD144) (138011, 1:100), anti-mouse CD45 (103112/103108, 1:100), anti-mouse CD3 (100214, 1:100), anti-mouse CD8 (100706, 1:100), anti-mouse NK1.1 (108714, 1:100), anti-mouse CD11b (101224, 1:100), and anti-mouse F4/80 (123116, 1:100) antibodies were purchased from Biolegend. InVivoMab anti-mouse PD-L1 (BE0101), anti-mouse CD8 (BE0004), and IgG2a isotype (BE0089) antibodies were purchased from BioXCell.

Mouse IL-2 (212-12) was purchased from Peprotech. Recombinant human IL-2 (BT-002-AFL) was purchased from R&D systems. L-ascorbate (VC) (A4034), NAC (A9165), collagenase IV (C5138), and DNase I (D5025) were purchased from Sigma. Fludarabine (S1491) was purchased from Selleck. 2′3′-cGAMP (HY-100564), TFMB-(R)-2-HG (HY-129079), STING inhibitor C-176 (HY-112906), and Sorafenib (Sora) (HY-10201) were purchased from MCE. MitoSOX Red mitochondrial superoxide indicator (M36008) was purchased from Invitrogen.

### Mice
$Cgas$ knockout mice ($Cgas^{-/-}$) (stock number: 026554) and $Sting$ knockout mice ($Sting^{-/-}$) (stock number: 025805), both on C57BL/6 background, were purchased from Jackson Laboratory. C57BL/6 mice were used as wild-type (WT) mice in this study and obtained from Laboratory Animal Resources, Chinese Academy of Sciences (Shanghai, China). Male mice at 5-6 weeks of age used in the experiments were obtained from the GemPharmatech Co., Ltd (Jiangsu, China). Mice were housed under 12 light/12 dark cycle, temperatures of $22 \pm 2$ °C with $50 \pm 10$ % humidity.

### Animal models
To establish subcutaneous tumor models, $1 \times 10^6$ murine liver cancer cells (Hepa1-6) or human liver cancer cells (Huh7) were implanted subcutaneously into the left thighs of male C57BL/6 mice or Nude mice at 5-6 weeks of age, respectively. Tumor growth was monitored by measuring the tumor size (length × width$^2$ × 0.5) twice per week after implantation. The permitted maximal tumor size did not exceed 1.5 cm at the largest diameter. In some cases, this limit has been exceeded the last day of measurement and the mice were immediately euthanized. At 3–4 weeks following tumor establishment, all the mice were sacrificed to harvest subcutaneous tumors for weight measurement and further tissue analyses. To establish orthotopic tumor models, laparotomy was administered to implant Hepa1-6 cells carrying a luciferase reporter gene into the left lobe of the liver after the mice were anaesthetized.

For bioluminescence imaging, mice were intraperitoneally injected with D-luciferin. After 15-20 min of injection, mice were subjected to imaging using IVIS Lumina III (PerkinElmer) and bioluminescence was quantified using Living Image software. Tumor volume was determined on the basis of the total flux (photons per second).

### In vivo treatments and depletion of CD8$^+$ T cells
Prior to treatments initiation, mice were randomly assigned to different groups with similar average tumor volumes. For in vivo treatments, VC (4 g/kg body weight; daily; i.p.) (Sigma) or PBS control, Sora (15 mg/kg body weight; daily; i.g.) (MCE); PD-L1 antibodies (75 μg per mouse; twice a week; i.p.) (BioXCell) or IgG isotype control, rIL-2 (1 μg per mouse; daily; i.p.) (R&D systems) or PBS control, and C-176 (750 nmol per mouse; daily; i.p.) (MCE) or solvent were injected for 2 weeks beginning on day 7 after the establishment of mice liver cancer models. To deplete CD8$^+$ T cells in vivo, mice were intraperitoneally injected with 100 μg of anti-CD8 antibody (BioXCell) or IgG isotype control (BioXCell) 3 days and 1 day before tumor implantation and twice weekly thereafter to ensure sustained depletion of CD8$^+$ T cell subset during the experimental period.

### In vivo vascular permeability assay
Tumor-bearing mice were injected intravenously with 0.5 mg of Evans blue dye (Sigma) in PBS. After 1 h of circulation, the mice were perfused through the heart with PBS containing 2 mM EDTA to remove intravascular Evans blue dye. The tumors were collected, cut into pieces, and incubated in 1 ml formamide for 24 h at 56 °C to extract the Evans blue from the tumor. After centrifugation, the supernatant was collected and diluted to a final concentration of same tumor weight per ml formamide. Then, the absorbances at 655 nm and 750 nm were measured with a spectrophotometer. To eliminate the effect of residual haem pigment in the blood, the following formula was used: corrected absorbance (A) 620 nm = A655 nm − (1.426 × A750 nm + 0.03) (7).

### siRNA, shRNA, CRISPR/Cas9 knockout, and overexpression
siRNAs used to knock down murine $Lrrc8c$, $Stat1$, and $Stat5a$ were conducted by Biotend (Shanghai, China). siRNAs were transfected into cells using Lipofectamine 2000 (Invitrogen) following the manufacturer's instructions. The Lentivirus-mediated shRNA expressing vectors targeting murine $Tet2$ (target sequence: 5′-GCTCTAAAT-GATGTAGCTTTG-3′) and $Stat5a$ (target sequence: 5′-GACGTGA-GATTCAAGTCTAAC-3′) were purchased from Genomeditech (Shanghai, China) and Genechem (Shanghai, China), respectively. Lentivirus-mediated CRISPR/Cas9 knockout (KO) vector targeting murine $Sting$ (target sequence: 5′-CAGCCTGATGATCCTTTGGG-3′) and $Cgas$ (target sequence: 5′-CGGCGGGCAGCTCCGGATCC-3′) was purchased from Obio Technology (Shanghai, China). The lentiviral plasmid expressing murine $Cgas$ was conducted by Biotend (Shanghai, China). The lentiviral plasmids expressing human $cGAS$, $STING$, and murine $Sting$-Cherry were purchased from Genechem (Shanghai, China). Cells (30% confluency) were transfected with optimal dilutions of lentivirus mixed with polybrene. Then, the transfected cells were treated with puromycin to select for stable transfected cells. Desired gene disruption was confirmed by immunoblot analysis of target proteins.

### Western blotting and co-immunoprecipitation (co-IP)
Whole cell lysates were prepared with cell lysis buffer (Beyotime Biotechnology) and boiled in SDS sample loading buffer. Then, an equal protein content (30 μg of protein per lane) was loaded to 10% PAGE electrophoresis and transferred onto PVDF membranes. The membranes were blocked in 5% milk in TBST for 1 h at room temperature and subsequently incubated overnight with the appropriate primary antibodies at 4 °C. After incubation with fluorescein-conjugated secondary antibody for 1 h at room temperature, the immunoblots were visualized using an Odyssey fluorescence scanner (Li-Cor, Lincoln, NE, USA). For co-IP, cells lysates (500 μg of protein) were mixed with 2 μg anti-Tet2 (GeneTex) overnight at 4 °C and then pulled down with protein G magnetic bead (Invitrogen) at 4 °C for 2 h. IP beads were then washed with lysis buffer three times, and heated in SDS sample loading buffer at 100 °C for 5 min for immunoblot analysis.

### Quantitative real-time PCR (qRT-PCR)
Total RNA was extracted from cells using TRIzol reagent (Invitrogen) and reversely transcribed into cDNA with random primers using Superscript III reverse transcriptase (Invitrogen). The cDNA was subsequently used as the template for the qRT-PCR reaction. qRT-PCR was performed using SYBR Green PCR Master Mix (Applied Biosystems) on ABI PRISM 7300HT Sequence Detection System (Applied Biosystems). β-Actin was used as a control for normalization. The primers were listed in Supplementary Table 1.

### Chromatin Immunoprecipitation (ChIP)
ChIP was performed by using EpiQuik Chromatin Immunoprecipitation (ChIP) Kit (Epigentek) according to the manufacturer's

instructions. In brief, cells were crosslinked and chromatin was extracted and sheared. The samples were immunoprecipitated with anti-Flag antibody (Abcam). qRT-PCR was performed on the immunoprecipitated DNA using SYBR Green PCR Master Mix (Applied Biosystems) on ABI PRISM 7300HT Sequence Detection System (Applied Biosystems). The values for immunoprecipitated DNA were normalized to input signals. The primers of *Cgas* promoter were listed in Supplementary Table 1.

### Flow cytometric analysis
Mice tumors were mechanically minced and incubated in collagenase IV (2 mg/mL, Sigma) and DNase I (50 μg/mL, Sigma) for 30 min at 37 °C with shaking. The dissociated cells were filtered through a 70 μm cell strainer (BD). Then, the resulting single-cell suspensions were incubated with Fc block and stained with the indicated surface antibodies for 20 min at 37 °C. The stained cells were analyzed immediately by a LSRFortessa flow cytometer (BD Biosciences) and analyzed using FlowJo software.

### cGAMP measurement
cGAMP levels in cell lysates and culture medium were measured using a 2′3′-cGAMP ELISA Kit (Cayman) according to manufacturer's protocol. For sample preparation, cells were lysed in M-PER™ Mammalian Protein Extraction Reagent (ThermoFisher Scientific).

### Lymphocytes transendothelial migration assay
Lymphocytes migration across an endothelial barrier in vitro was assessed using CytoSelect™ Leukocyte Transmigration Assay kit (Cell Biolabs) according to the manufacturers' instruction. Lymphocytes used in the study were isolated from the spleens of WT or *Sting*$^{-/-}$ mice with Mouse Lymphocyte Separation Medium (Dakewe Biotechnology) following manufacturer's protocol, and then labeled with LeukoTracker™. The endothelial monolayers were exposed to indicated treatment for 24 h before placing the LeukoTracker™ labeled lymphocytes on the endothelial monolayer. The relative abundance of transendothelial migrated lymphocytes was calculated by measuring the fluorescence of the samples at 480 nm/520 nm.

### Immunofluorescence staining
Frozen mouse tumor sections or cells were fixed with 4% paraformaldehyde for 20 min at room temperature and were permeabilized with 1% Triton X-100 (Solarbio) solution for 15 min. Next, 5% BSA in PBS with 0.1% triton was used to block non-specific binding sites at room temperature for 1 h. Tumor sections were stained with indicated primary antibodies against CD31 and α-SMA, and cells were stained with primary antibodies against dsDNA and HSP60 overnight at 4 °C followed by fluorescent secondary antibodies at room temperature for 1 h. DAPI (Life Technologies) was used for nuclear staining. All the slides were visualized and photo-documented using a STELLARIS 5 confocal microscope (Leica Microsystems).

### Endothelial cell proliferation and migration assay
SVEC4-10 cells were seeded at a density of 3000–5000 cells per well in a 96-well plate. After indicated treatment for 24 h, cell viability was measured using Cell Counting Kit-8 (CCK-8) assay (Vazyme) following manufacturer's instruction. For migration assay, 5 × 10$^4$ overnight serum-starved SVEC4-10 cells were seeded on the PET membrane (8 μm pore size) of an insert. The inserts were hanged on a 24-well support plate, which had a 70–80% confluent monolayer of cultured Hepa1-6-Ctrl or Hepa1-6-*Cgas* cells with complete growth media. After 24 h of incubation, the inserts were fixed with 4% paraformaldehyde and stained with crystal violet solution. Representative fields were photographed and the migrated cells were quantified by ImageJ software (NIH).

### Tube formation assay
SVEC4-10 cells were exposed to indicated treatment for 24 h, and subsequently plated on Matrigel (Corning) coated 24-well plates (3 × 10$^4$ cells/well) for 6 h at 37 °C before taking pictures. Angiogenesis analyzer module of ImageJ software (NIH) was applied to quantify the number of meshes and total tube length.

### Immunohistochemical (IHC) staining
IHC staining was performed on representative tissue sections from formalin-fixed and paraffin-embedded tissue blocks from human HCC and mice tumor xenografts using the indicated antibodies. Based on the immunoreactive score method, the intensity of human HCC tissue microarrays staining (protein expression) was scored as 0 (negative staining), 1 (weak staining), 2 (moderate staining), and 3 (strong staining). Then, the patients were subdivided into two groups: low expression group (negative or weak staining) and high expression group (moderate or strong staining). For CD31/α-SMA dual IHC staining, CD31 was stained first with 3,3-diaminobenzidine (DAB) (brown chromogen) and α-SMA was selected second to stain with Cynanine3 (red chromogen). Pericyte coverage of tumor vessels was estimated by measuring the vessels which stained positively for both CD31 and α-SMA (pericytes marker) among all CD31-positive vessels. The results were shown as the percentage of tumor blood vessels with α-SMA-positive pericyte coverage.

### DNA methylation pyrosequencing
*Cgas* methylation levels were measured using pyrosequencing to examine the methylation status at CpG sites in the *Cgas* promoter. The sequencing service and bioinformatics analysis were provided by Shanghai Biotechnology Corporation (Shanghai, China). Briefly, genomic DNA was extracted from cells and then the DNA samples were subjected to bisulfite conversion using the EZ DNA Methylation Kit (Zymo Research) according to the manufacturer's instructions. Bisulfite-treated DNA was used as template to amplify the target fragment of *Cgas* promoter with the specific primer pair. PCR products were sequenced by pyrosequencing using PyroMark Q96 (QIAGEN).

### Online database analysis
cGAS (C6orf150, MB21D1) and STING (TMEM173) protein expressions in liver cancer tissues assessed by immunohistochemistry were obtained from the Human Protein Atlas (HPA, http://www.proteinatlas.org)[61]. The correlations between different genes expressions in HCC were obtained from The Cancer Genome Atlas (TCGA, https://portal.gdc.cancer.gov) and the International Cancer Genome Consortium (ICGC, https://dcc.icgc.org) databases. CD8$^+$ T cell infiltration analysis according to different genes expressions in HCC from TCGA database was conducted using Tumor Immune Estimation Resource (TIMER)[62]. The DNA methylation data from TCGA were derived from DNA methylation interactive visualization database (DNMIVD)[63]. Gene Set Enrichment Analysis (GSEA) and Panther Pathway Analysis in HCC TCGA datasets was performed in the LinkedOmics platform[64]. The expressions of IL-2 receptors in human liver cancer cell lines were obtained from the Cancer Cell Line Encyclopedia (CCLE, https://sites.broadinstitute.org/ccle). Survival differences were validated at the gene expression level in HCC TCGA by an online database Kaplan-Meier Plotter (KM-Plotter)[65].

### Statistics
Statistical analysis was performed using GraphPad Prism 8 software (GraphPad Software, San Diego, CA). Data are presented as mean ± standard error (SD) unless otherwise stated. Experiments were repeated at least three times with similar results. The specific statistical tests applied are given in the respective figure legends. $P$ value < 0.05 was considered to be statistically significant.

**Reporting summary**

Further information on research design is available in the Nature Portfolio Reporting Summary linked to this article.

## Data availability

Publicly available datasets reported in this paper are from the GEO databases (GSE51401, GSE69164, GSE146409, GSE140901), The Cancer Genome Atlas (TCGA, https://portal.gdc.cancer.gov), the International Cancer Genome Consortium (ICGC, https://dcc.icgc.org), the Human Protein Atlas (HPA, http://www.proteinatlas.org), and the Cancer Cell Line Encyclopedia (CCLE, https://sites.broadinstitute.org/ccle). The remaining data are available within the Article, Supplementary Information or Source Data file. Source data are provided with this paper.

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

## Acknowledgements

We thank Dr. Wuhan Xiao and Dr. Guoliang Xu (Chinese Academy of Sciences) for Tet plasmids, and Dan Cao, Shanhua Tang, Linna Guo, Lu Chen, Mingshuang Xu, Shennian Ge, and Qi Yang for their technical assistance. This work was supported by grants from the National Natural Science Foundation of China (82273176, 81902894, 81972779, 81903036, 81830054, 91859205, and 81988101), Chinese National Key Project (2018ZX10723204-006-003), Shanghai Municipal Commission of Education Project (201901070007E00065), and Program of Shanghai Academic Research Leader (23XD1404800).

## Author contributions

W.Y., H.L., and H.W. designed the study. H.L., Q.Z., C.C., W.X., F.X., and G.L. performed the experiments and data analysis. G.J, B.Y., X.S., Y.M., L.W., Z.W., and X.C. provided technical support. H.L. and W.Y. wrote the manuscript. H.W. and W.Y. organized and supervised the study.

## Competing interests

The authors declare no competing interests.
