## [Peer Review File · Nature Communications]

TET2-mediated tumor cGAS triggers endothelial STING activation to regulate vasculature remodeling and anti-tumor immunityREVIEWER COMMENTS

Reviewer #1 (Remarks to the Author): with expertise in cancer immunology, cGAS/STING

In this manuscript, Lv and colleagues investigate the role of tumor-derived 2'3'-cGAMP produced by tumoral cGAS in the tumor microenvironment (TME). They demonstrate that tumoral cGAS is secreted via LRRC8C and stimulates STING in endothelial cells, leading to vascular normalization and lymphocyte transmigration. The authors report that tumoral cGAS is often suppressed by DNA hypermethylation and attempt to reactivate it using vitamin C (VC) treatment. VC is shown to upregulate the TET2/STAT5a complex, inducing DNA demethylation in the cGAS promoter region.

The study employs a comprehensive approach, using cell lines, mouse models, and clinical samples to support the conclusions. The authors broadly investigate the interaction between tumor cells and the TME, providing insight into the regulation of the tumoral cGAS and its effect on vascular normalization and sensitivity to immune checkpoint inhibitors. While the study is methodologically sound and provides a valuable contribution to the field, I think it lacks conceptual novelty for a journal such as Nature Communications. Previous work has already reported tumor-derived 2'3'-cGAMP secretion and stimulation of the STING pathway in vascular endothelial cells, leading to ICAM up-regulation (PMID:33013881). The essential role of tumoral cGAS and host STING for the activation of STING signaling in the TME has also been established (PMID:30332631, 31665636, etc.), although the authors mention these papers in the introduction. The proposed model of TET2/STAT5a directly regulating tumoral cGAS expression is intriguing. Additional experiments could strengthen the underlying mechanisms, particularly by exploring the effects of more specific agents instead of using non-specific agents such as VC. In conclusion, the manuscript by Lv and colleagues presents a well-executed and thorough investigation into the role of tumoral cGAS to regulate vasculature remodeling in the TME. To elevate the conceptual novelty and potential impact for a journal like Nature Communications, further experiments and a more detailed discussion of the novel aspects of the study are recommended. We provide the following constructive suggestions:

1. Throughout the paper, the authors primarily use VC to activate tumoral cGAS through TET2-dependent demethylation of DNA in the cGAS promoter region. While the authors

have conducted various in vitro experiments to demonstrate this, VC may not be the most specific and potent drug to establish TET2 dependency conclusively. To further support their findings, it is suggested that the authors perform experiments using a combination of specific inhibitors or genetic depletion of TET2 (or STAT5a) alongside cGAS KO and STING inhibitor with VC in Fig. 7 and Sup.Fig.11. This will allow them to more accurately evaluate the impact of their findings on vascular normalization and the efficacy of immune checkpoint inhibitors.

2. The authors focused on the methyl-cytosine dioxygenase TET2 as a potential regulator of DNA methylation, based on the observed correlation between TET2 and cGAS expression. On the other hand, several studies have demonstrated the significant role of the DNMT family in the DNA methylation of the cGAS promoter. It would be valuable to investigate if treatment with clinically relevant DNMT inhibitors, such as azacytidine and decitabine, also leads to upregulation of tumor cGAS and subsequent 2'3'-cGAMP-dependent vascular normalization in their model. This comparison could help contextualize their findings within established molecular mechanisms in the field.

3. In Figure 1, the authors compare the effects of cGAS overexpression and the loss of baseline STING expression to determine whether tumoral STING and its downstream targets play a role in vascular normalization. However, STING expression alone does not necessarily indicate its activation. It may be premature to dismiss the potential function of tumoral STING in this context. In fact, several studies have reported that IFN- β , a crucial downstream cytokine of STING, also up-regulates ICAM expression in vascular endothelial cells (PMID:25217157, 33013881). To improve the analysis in Figure 1, the authors should include an additional condition: cGAS overexpression in STING-positive cancer cells. By comparing this to cGAS overexpression in a STING-null background, they can accurately assess STING's potential role in vascular normalization. Additionally, the authors should overexpress STING in Huh7 cells and compare the resulting phenotype to that of cGAS-reconstituted cells in Supplementary Figure 3. This would further refine the understanding of tumoral STING's involvement in the process

4. The molecular mechanisms underlying how VC treatment triggers the export of damaged

DNA fragments derived from the nucleus and/or mitochondria remain unclear. Is this phenomenon specific to tumor cells? Given that ascorbic acid is a widely recognized antioxidant, providing more information and discussion on previous research in this area would be beneficial for many readers.

Reviewer #2 (Remarks to the Author): with expertise in HCC, (immuno)therapy

Hongwei Lv and colleagues suggest that HCC tumor cells expressing cGAS, release cGAMP through a dedicated ion channel (LRRC8C) which is sensed solely by endothelial cells. This causes blood vessel maturation which allegedly enhances the ability of the immune system to counterfeit tumor growth. They identify high-dose vitamin C as a compound that can specifically upregulate expression of TET2 (not mediated by VC's ability to cause H₂O₂ stress), that results in upregulation of cGAS, thus potentiating anti tumor immune responses to immune checkpoint blockers (ICBs).

Major comments:

1. The key results are generated from cells with marked overexpression of cGAS (Fig. 1A). Thus it is possible that all the observed effects are an artifact that is not related to endogenous tumor cell biology. Are the levels of Cgas and cGAMP that are achieved in the transfected cells related to endogenous levels in human HCC? Indeed, in their TET studies, they show that overexpressing TET raises the intracellular levels of cGAMP by 1.5 fold to about 150 pg/mg protein (Fig. 4g), but in their cGAS overexpressing cells, the cGAMP is increased to 1500 pg/mg protein. Looking at fig. 7N – they actually did this experiment and it seems that in strike contrast to their major claim, tumors derived from cGAS KO Hepa1-6 cells are much smaller than their cGAS wt counterparts. How can this be explained?
2. They assume that the normalized blood vessels are the cause of the difference in tumor growth, but it could be that blood vessels in slower-growing tumors have more time to mature. This must be tested.
3. It is highly unlikely that VC is a specific regulator of TET2 expression. There are multiple other explanations for its effects on ICB therapy. A transcriptome study will likely reveal multiple other targets of VC, and it's important to see if TET2 is one of the most upregulated genes in this scenario.

4. There are quite a lot of studies reporting effects of the cGAS-Sting pathway in HCC, in which effects were noted in the malignant cells, in dendritic cells and in tumor-associated macrophages. So it's unclear to me why all these are not seen in the current model.

Minor comments

1. The figures are not referred to in their order (S1b after S2, etc.).
2. Liver cancer is not the leading cause of cancer-related death globally
3. Figure 1 shows that Cgas expression in tumor cells, leads to reduced tumor growth, which depends on host Sting. They should also test Cgas- cells (i.e. native cells), in Sting wt and KO mice, as it is possible that the effect seen in fig. 1g does not depend on the overexpression of Cgas.
4. A critical control that is missing throughout the manuscript are Cgas+Sting+ cells.
5. It seems that all the experiments are done on single clones. Thus, at least some key experiments must be repeated in different clones (particularly Cgas overexpression).
6. Fig S1b shows a critical control. The in vitro testing should be expanded to all the clones, not just the two shown.
7. The authors state that adding anti-angiogenic therapy to immune checkpoint blockade synergizes with the latter as it induces 'vascular normalization'. While this has been argued there are many other mechanisms, most importantly a direct role of VEGF as a potent immunomodulatory agent, particularly of tumor-associated macrophages.
8. The text referring to fig. 1g is wrong. The cells were not rejected in the wt mice, they grew slower. This could be related to many reasons, other than rejection.
9. The conditioned media experiment is not clear to me as the half-life of cGAMP is thought to be minutes. It is important to measure the levels of cGAMP in the media when the cells are treated.
10. Fig. 4C – 5HmC is not a demethylation marker, it is an intermediate in the pathway. Once a methylated cytosine is demethylated it is no longer detectable as there is no need for ongoing demethylation. So the correlation they detected does not make sense. Moreover, the staining seems to be cytoplasmic, while TET enzymes catalyze the hydroxylation of DNA-5-methylcytosine (5mC), not free cytosine.
11. The 2HG they used (disodium (R)-2-Hydroxyglutarate (Catalog No. S7873)) is not cell permeable. So how did it inhibit TET enzymes? (Fig. 5G)
12. The immunostain for LRRC8C is cytoplasmic, while it should be membranous. So this is

likely an artifact.

13. Fig. 7k-m – looking at the graphs, it looks like there is a significant effect of both VC alone and aPDL1 alone in the CD8 depleted mice. The magnitude of this effect is similar to that in the IgG control group. There is a single outlier in the combined group, which led them to conclude that there is no effect of the combination. I don't understand how the aPDL1 works in CD8 depleted mice.

14. Fig. 7n-p. it is clear that both aPD-L1 and VC affect growth of cGAS KO tumors; moreover, the magnitude of their effect as monodrugs, are higher in the cGAS KO tumors (particularly that of aPD-L1). Doesn't that mean that cGAS expression impedes the efficacy of ICB therapy (or at least doesn't enhance it)?

Reviewer #3 (Remarks to the Author): with expertise in epigenetics, immunology

In this paper, Hongwei and colleagues elucidate the crosstalk between tumor and vascular endothelial cells in the tumor microenvironment via cGAS and STING. They define how the expression of cGAMP by hepatocellular carcinoma cells is used to activate STING in endothelial cells and this promotes the recruitment of T cells that transmigrate and produce there IL-2. IL-2 activates STAT5 which recruits TET2 to specifically demethylate and promote the expression of cGAMP that again induces tumor vascular normalization. Authors propose Vitamin C as a potential immunotherapy as this can enhance the activity of TET2, improve vascular normalization and improve response to anti-PD-L1 therapy.

The study is well conducted and the results are sound. I have no additional comments and the paper is acceptable for publication.

Reviewer #4 (Remarks to the Author): with expertise in STING, cancer immunology, vascular normalization

Lv and colleagues report that tumor cell expressed TET2 epigenetically regulates intrinsic cGAS activity, resulting in local STING activation (in tumor-associated VEC leading to vascular normalization) and a proinflammatory TME conducive to effective treatment with ICI (i.e., anti-PD-L1). Vitamin C activates TET2 activity and synergizes with anti-PD-L1 (in a tumor

cGAS- and host STING-dependent manner) in effectively treating established (predominantly s.c., with 1 experiment in orthotopic) liver carcinoma models in C57BL/6 mice.

The main aspect of novelty for the current report stems from its identification of tumor cGAS as a TET2 downstream target that regulates cGAMP production/secretion and local STING activation in host cells (VEC) within the TME to regulate tumor progression, with both cGAMP export (from tumor cells) and import (by VEC) being dependent on LRRC8 channels. Others have previously shown: 1.) that tumor TET2 activity (constitutive or vitamin C-induced) mediates IFNG-JAK-STAT signaling thereby regulating, local pro-inflammatory chemokine production and PD-L1 expression, TIL recruitment and response to anti-PD-L1 in B16 melanoma and MC38 colon carcinoma models in mice while serving as a putative biomarker for predicting patient response to ICI (PMID 31310587, not cited in the current report) and that 2.) host STING activation promotes vascular normalization, improved TIL recruitment and enhanced response to ICI (including anti-PD-L1, several publications cited by the authors).

Overall, the studies appear well-performed and well-controlled, and the conclusions reached by the authors are appropriately justified. There are some areas where the report can be improved:

1.) Although the authors would like to model the impact of their operating paradigm in highly-vascularized liver cancer, the mouse model employs predominantly subcutaneous (s.c.) models with 1 instance of an orthotopic model (in Fig. 7i/j), which limits translational conclusions from being made relevant to human disease. Differences in s.c. vs. orthotopic models is not compared/contrasted, although one would expect major differences in baseline immune status between skin (pro-inflammatory) vs. liver (anti-inflammatory) that could impact mechanism of action. Inclusion of such information would improve the merit of the report.

2.) Per the animal modeling data (Fig. 7), studies are very short term in nature (< 20 days), making it difficult to discern treatment impact on animal survival. Kaplan-Meier plots and statistical comparisons between cohorts should be provided by the authors in a revised report.

3.) Further to the animal modeling, given the authors' indication that IL-2R/STAT5 signaling contributes to tumor cGAS activity, there is an opportunity lost in the failure to test rIL-2 along with VC +/- anti-PD-L1 for enhanced therapeutic potential in s.c. and orthotopic HCC models.

4.) (minor) The report cites 80 references which seems excessive for a non-review article. The authors should reduce this number by focusing on only the most critical supportive citations.

Response to reviewers' comments

Reviewer #1 (Remarks to the Author): with expertise in cancer immunology, cGAS/STING

In this manuscript, Lv and colleagues investigate the role of tumor-derived 2'3'-cGAMP produced by tumoral cGAS in the tumor microenvironment (TME). They demonstrate that tumoral cGAS is secreted via LRRC8C and stimulates STING in endothelial cells, leading to vascular normalization and lymphocyte transmigration. The authors report that tumoral cGAS is often suppressed by DNA hypermethylation and attempt to reactivate it using vitamin C (VC) treatment. VC is shown to upregulate the TET2/STAT5a complex, inducing DNA demethylation in the cGAS promoter region.

The study employs a comprehensive approach, using cell lines, mouse models, and clinical samples to support the conclusions. The authors broadly investigate the interaction between tumor cells and the TME, providing insight into the regulation of the tumoral cGAS and its effect on vascular normalization and sensitivity to immune checkpoint inhibitors. While the study is methodologically sound and provides a valuable contribution to the field, I think it lacks conceptual novelty for a journal such as Nature Communications. Previous work has already reported tumor-derived 2'3'-cGAMP secretion and stimulation of the STING pathway in vascular endothelial cells, leading to ICAM up-regulation (PMID:33013881). The essential role of tumoral cGAS and host STING for the activation of STING signaling in the TME has also been established (PMID:30332631, 31665636, etc.), although the authors mention these papers in the introduction. The proposed model of TET2/STAT5a directly regulating tumoral cGAS expression is intriguing. Additional experiments could strengthen the underlying mechanisms, particularly by exploring the effects of more specific agents instead of using non-specific agents such as VC. In conclusion, the manuscript by Lv and colleagues presents a well-executed and thorough investigation into the role of tumoral cGAS to regulate vasculature remodeling in the TME. To elevate the conceptual novelty and potential impact for a journal like Nature Communications, further experiments and a more detailed discussion of the novel aspects of the study are recommended. We provide the following constructive suggestions:

1. Throughout the paper, the authors primarily use VC to activate tumoral cGAS through TET2-dependent demethylation of DNA in the cGAS promoter region. While the authors have conducted various in vitro experiments to demonstrate this, VC may not be the most specific and potent drug to establish TET2 dependency conclusively. To further support their findings, it is suggested that the authors perform experiments using a combination of specific inhibitors or genetic depletion of TET2 (or STAT5a)

alongside cGAS KO and STING inhibitor with VC in Fig. 7 and Sup.Fig.11. This will allow them to more accurately evaluate the impact of their findings on vascular normalization and the efficacy of immune checkpoint inhibitors.

We appreciate the reviewer's constructive comments. As requested, we knocked down Tet2 or Stat5a expression in Hepa1-6 cells. Subsequently, we observed that in tumors derived from Tet2 or Stat5a deficiency cells, the combination of VC and anti-PD-L1 had no obvious changes in tumor burdens as compared to single VC or anti-PD-L1 therapy (**Fig. 9a-d**). Moreover, VC therapy did not exert a significant impact on tumor vessel coverage by pericytes, intratumoral hypoxia levels, and CD8⁺ T cell infiltration in these tumors with Tet2 or Stat5a deficiency (**Fig. 9e, Supplementary Fig. 12g**). Furthermore, the knockdown of Tet2 or Stat5a in tumors negated the upregulation induced by VC treatment of IFN β and ISGs, vascular stabilizing genes, endothelial-lymphocyte interaction-associated adhesion molecules (**Supplementary Fig. 12h**). Collectively, these findings provide conclusive evidence for the crucial involvement of tumor Tet2 and Stat5a in VC-mediated vascular normalizing effects and the combinational efficiency of VC and anti-PD-L1 therapy in liver cancer.

2. The authors focused on the methyl-cytosine dioxygenase TET2 as a potential regulator of DNA methylation, based on the observed correlation between TET2 and cGAS expression. On the other hand, several studies have demonstrated the significant role of the DNMT family in the DNA methylation of the cGAS promoter. It would be valuable to investigate if treatment with clinically relevant DNMT inhibitors, such as azacytidine and decitabine, also leads to upregulation of tumor cGAS and subsequent 2'3'-cGAMP-dependent vascular normalization in their model. This comparison could help contextualize their findings within established molecular mechanisms in the field.

We thank the reviewer for raising this important point. As suggested, decitabine, a DNMT inhibitor was employed. We observed that treatment with various concentrations of decitabine did not elicit a noticeable impact on Cgas expression in Hepa1-6 cells, as shown in the following graph. Additionally, our analysis of HCC samples from TCGA and ICGC datasets revealed no significant negative correlation between tumor cGAS expression and the DNMT family (**Fig. 5c**). Hence, further investigation is needed to clarify whether the DNMT family plays a predominant role in influencing the DNA methylation status of the cGAS promoter in our model. This issue has been addressed in our revised manuscript.

3. In Figure 1, the authors compare the effects of cGAS overexpression and the loss of baseline STING expression to determine whether tumoral STING and its downstream targets play a role in vascular normalization. However, STING expression alone does not necessarily indicate its activation. It may be premature to dismiss the potential function of tumoral STING in this context. In fact, several studies have reported that IFN- β , a crucial downstream cytokine of STING, also up-regulates ICAM expression in vascular endothelial cells (PMID:25217157, 33013881). To improve the analysis in Figure 1, the authors should include an additional condition: cGAS overexpression in STING-positive cancer cells. By comparing this to cGAS overexpression in a STING-null background, they can accurately assess STING's potential role in vascular normalization. Additionally, the authors should overexpress STING in Huh7 cells and compare the resulting phenotype to that of cGAS-reconstituted cells in Supplementary Figure 3. This would further refine the understanding of tumoral STING's involvement in the process

Thanks to the reviewer's constructive comment. Per the reviewer's suggestion, we knocked out *Sting* in *Cgas* overexpression cells to further assess the role of tumoral STING in tumor vascular normalization and anti-tumor immunity. As shown in **Fig. 1g**, knockout of *Sting* in Hepa1-6 cells (*Cgas* (+)/*Sting* (-)) did not alter tumor growth compared to parental cells (*Cgas* (+)/*Sting* (+)) (**Fig. 1h, i**). Moreover, no obvious differences were observed in α -SMA⁺ pericyte coverage of CD31⁺ tumor vessels, or intratumoral T cell infiltration (**Fig. 1j**), indicating that tumoral STING deficiency had little effect on vascular normalization and anti-tumor immunity. Likewise, STING overexpression in cGAS-reconstituted Huh7 cells developed comparable tumor sizes to parental STING-null cells and no significant difference in tumor vessel coverage by pericytes was found after STING overexpression (**Supplementary Fig. 3f-i**). Collectively, these data indicate that tumor-intrinsic STING expression is dispensable for vascular normalization and anti-tumor immunity in the context of an intact tumor-intrinsic cGAS background.

4. The molecular mechanisms underlying how VC treatment triggers the export of damaged DNA fragments derived from the nucleus and/or mitochondria remain unclear. Is this phenomenon specific to tumor cells? Given that ascorbic acid is a widely recognized antioxidant, providing more information and discussion on previous research in this area would be beneficial for many readers.

We appreciate the reviewer's concern. Numerous studies have demonstrated that pharmacological VC at millimolar concentrations can kill cancer cells *in vitro* and *in vivo* (1). Nevertheless, the mechanism by which some cancer cells are sensitive to VC, while normal cells remain resistant, is remains inadequately elucidated. The selective toxicity of VC in cancer cells could depend on a variety of different factors, including redox imbalance, iron level, epigenetic regulators (1). There are various mechanisms that trigger the export of damaged DNA fragments derived from the nucleus and/or mitochondria, such as formation of abnormal micronucleus, mitochondrial degeneration and membrane potential reduction, autophagy (2,3). Undoubtedly, further studies are needed to clarify the mechanisms underlying how VC treatment triggers the export of damaged DNA fragments derived from the nucleus and/or mitochondria, and to ascertain the specificity of this modulation towards tumor cells. This issue has been addressed in the revised manuscript.

Reviewer #2 (Remarks to the Author): with expertise in HCC, (immuno)therapy

Hongwei Lv and colleagues suggest that HCC tumor cells expressing cGAS, release cGAMP through a dedicated ion channel (LRRC8C) which is sensed solely by endothelial cells. This causes blood vessel maturation which allegedly enhances the ability of the immune system to counterfeit tumor growth. They identify high-dose vitamin C as a compound that can specifically upregulate expression of TET2 (not mediated by VC's ability to cause H₂O₂ stress), that results in upregulation of cGAS, thus potentiating anti tumor immune responses to immune checkpoint blockers (ICBs).

Major comments:

1. The key results are generated from cells with marked overexpression of cGAS (Fig. 1A). Thus it is possible that all the observed effects are an artifact that is not related to endogenous tumor cell biology. Are the levels of Cgas and cGAMP that are achieved in the transfected cells related to endogenous levels in human HCC? Indeed, in their TET studies, they show that overexpressing TET raises the intracellular levels of cGAMP by 1.5 fold to about 150 pg/mg protein (Fig. 4g), but in their cGAS overexpressing cells, the cGAMP is increased to 1500 pg/mg protein.

We thank the reviewer for raising this important issue. To mimic a more faithful emulation of the endogenous cGAS/STING pathway levels observed in human hepatocellular carcinoma (HCC), a series of distinct clones with differential Cgas expression levels were selected (**Fig. 1k**). As shown in **Fig. 1l**, we observed a broader range of intracellular and extracellular cGAMP concentrations, spanning from a few hundred pg/mg protein to tens of thousands pg/mg protein. Importantly, we found that with higher Cgas expression in cancer cells, smaller tumor burdens were observed (**Fig. 1m, n**), while more pericyte coverage of tumor vessels and intratumoral T cell infiltration were detected (**Fig. 1o**), suggesting that tumor cGAS mediates tumor repression, vascular normalization, and anti-tumor immune response in a cGAS expression level-dependent manner.

Looking at fig. 7N – they actually did this experiment and it seems that in strike contrast to their major claim, tumors derived from cGAS KO Hepa1-6 cells are much smaller than their cGAS wt counterparts. How can this be explained?

We apologize that this point may not have been described clearly enough. The cGAS KO experiment and cGAS wt counterparts experiment were conducted as separate, independent experiments, so this comparison is not very rigorous or accurate. As endogenous Cgas expression level was very low in native Hepa1-6 cells (**Supplementary Fig. 1a**), Cgas KO in Hepa1-6 is not suitable to directly test the role of cGAS in tumor biology. Instead, Cgas was overexpressed in native Hepa1-6 cells to

assess the effect of tumor cGAS on tumor vasculature and immunity in our models. Nevertheless, to abrogate Cgas expression upregulated by VC treatment, we knocked out Cgas in Hepa1-6 cells to confirm whether VC treatment-induced tumor vascular normalization and anti-tumor immune response depends on cGAS expression.

2. They assume that the normalized blood vessels are the cause of the difference in tumor growth, but it could be that blood vessels in slower-growing tumors have more time to mature. This must be tested.

We thank the reviewer for raising this important point. To further evaluate the role of cancer cell-intrinsic cGAS and STING in tumor blood vessel maturation, we tested the expression status of VEGF receptor 2 (VEGFR2), a molecular marker highly expressed by the endothelium of growing, immature vessels (4). As shown in **Fig. 1f**, **Supplementary Fig. 2c**, tumor vessels were generally more mature in Cgas overexpression tumors, as evidenced by the majority of the vessels expressing low levels of VEGFR2. In contrast, cancer cell-intrinsic Sting overexpression exhibited no substantial effect on tumor blood vessel maturation with no difference in the proportion of VEGFR2-expressing vessels. Therefore, tumor cGAS promotes the maturation of tumor vessels, which is essential for functional and normalized vasculature.

3. It is highly unlikely that VC is a specific regulator of TET2 expression. There are multiple other explanations for its effects on ICB therapy. A transcriptome study will likely reveal multiple other targets of VC, and it's important to see if TET2 is one of the most upregulated genes in this scenario.

Thanks to the reviewer's constructive comment. It is well-established that VC potentiates TET enzyme activity by acting as a co-factor for Fe²⁺(+) and alpha-KG-dependent dioxygenases, rather than through the upregulation of TET expression (5). Thus, whether TET2 is the most upregulated genes by VC is not the principal objective of our study in this particular context.

In spite of VC's various other targets, our study revealed that Tet2 deficiency abrogated the synergistic anti-tumor effect of VC and PD-L1 blockade in vivo (**Fig. 9a-d**), as well as the VC-induced vascular normalizing effects (**Fig. 9e**, **Supplementary Fig. 12g, h**). Mechanistically, our findings indicate that VC upregulated tumor cGAS, further produced cGAMP to activate endothelial STING and enhance transendothelial migration of lymphocytes in a Tet2-dependent manner.

4. There are quite a lot of studies reporting effects of the cGAS-Sting pathway in HCC, in which effects were noted in the malignant cells, in dendritic cells and in tumor-associated macrophages. So it's unclear to me why all these are not seen in the current model.

We agree with the reviewer that the cGAS-STING pathway plays important roles in regulating tumor cells and immune cells, including dendritic cells, tumor-associated macrophages, and T cells. Although we revealed that VC-induced TET2 and dsDNA leakage activate tumor cGAS-cGAMP-endothelial STING pathway and promote lymphocyte trafficking, we can't exclude the possibility that the increase cGAMP in the tumor microenvironment can also impact the function of immune cells directly. Therefore, we hope to address this issue in future studies.

Minor comments

1. The figures are not referred to in their order (S1b after S2, etc.).

The figures have been corrected as requested.

2. Liver cancer is not the leading cause of cancer-related death globally

We apologize for this wrong description. According to the recent report (6), we have described liver cancer as the third leading cause of cancer-related death globally and have revised the text accordingly.

3. Figure 1 shows that Cgas expression in tumor cells, leads to reduced tumor growth, which depends on host Sting. They should also test Cgas- cells (i.e. native cells), in Sting wt and KO mice, as it is possible that the effect seen in fig. 1g does not depend on the overexpression of Cgas.

Thanks to the reviewer's constructive comment. We challenged *Sting*^{-/-} or WT mice with native ctrl cells with low Cgas expression and found that these cells developed similar tumors in *Sting*^{-/-} or WT mice (**Fig. 2g, h**). Furthermore, no obvious differences in vascular normalization and CD8⁺ T cells infiltration (**Fig. 2g, h**), as well as vascular normalizing genes expressions were observed between *Sting*^{-/-} and WT mice (**Supplementary Fig. 4d**). Furthermore, *Sting*^{-/-} mice implanted with Cgas-proficient cells developed comparable tumors than parental ctrl cells (**Fig. 2c, d**), and tumor Cgas-induced vascular normalization and increased CD8⁺ T cells infiltration were also abrogated in *Sting*^{-/-} mice (**Fig. 2k, Supplementary Fig. 4b**). Together, these results further indicate that tumor cGAS mediates vascular normalization and anti-tumor immune response in a host STING-dependent manner.

4. A critical control that is missing throughout the manuscript are Cgas+Sting+ cells.

We thank the reviewer for raising this important point. The missing of Cgas+Sting+ cells mainly impairs the evaluation of the role of tumor-intrinsic STING in vascular normalization and anti-tumor immunity under tumor-intrinsic cGAS overexpression background. Therefore, we knocked out Sting in Cgas overexpression Hepa1-6 cells to further assess the role of tumoral STING in tumor vascular normalization and anti-

tumor immunity. As shown in **Fig. 1g**, knockout of Sting in Hepa1-6 cells (Cgas (+)/Sting (-)) did not alter tumor growth compared to parental cells (Cgas (+)/Sting (+)) (**Fig. 1h, i**). Moreover, no obvious differences were observed in α -SMA⁺ pericyte coverage of CD31⁺ tumor vessels, or intratumoral T cell infiltration (**Fig. 1j**), indicating that tumoral STING deficiency had little effect on vascular normalization and anti-tumor immunity. Likewise, STING overexpression in cGAS-reconstituted Huh7 cells developed comparable tumor sizes to parental STING-null cells and no significant difference in tumor vessel coverage by pericytes was found after STING overexpression (**Supplementary Fig. 3f-i**). Together, these data suggest that tumor-intrinsic STING expression is not required for vascular normalization and anti-tumor immunity under intact tumor-intrinsic cGAS background.

5. It seems that all the experiments are done on single clones. Thus, at least some key experiments must be repeated in different clones (particularly Cgas overexpression).

We thank the reviewer for raising this important point. Since tumor cGAS-mediated vascular normalization and anti-tumor immune response is not only the vital basis of our story but also the rationale for further testing VC combined with anti-PD-L1 therapy, we selected a series of distinct clones with differential Cgas expression levels in liver cancer cells to confirm this important finding (**Fig. 1k**). As shown in **Fig. 1l**, with higher Cgas expression in cancer cells, smaller tumor burdens were observed (**Fig. 1m, n**), while more pericyte coverage of tumor vessels and intratumoral T cell infiltration were detected (**Fig. 1o**), suggesting that tumor cGAS mediates tumor repression, vascular normalization, and anti-tumor immune response in a cGAS expression level-dependent manner.

6. Fig S1b shows a critical control. The in vitro testing should be expanded to all the clones, not just the two shown.

Per the reviewer's suggestion, we tested cell proliferation of all the clones in vitro and confirmed that the cell proliferation rates were similar after Cgas overexpression in all the clones (**Supplementary Fig. 1b**).

7. The authors state that adding anti-angiogenic therapy to immune checkpoint blockade synergizes with the latter as it induces 'vascular normalization'. While this has been argued there are many other mechanisms, most importantly a direct role of VEGF as a potent immunomodulatory agent, particularly of tumor-associated macrophages.

We apologize for this less accurate and comprehensive description about the mechanism underlying the combination of anti-angiogenic therapy and immune checkpoint blockade. Anti-angiogenic therapy can not only normalize aberrant tumor

blood vessels to improve intratumoral immune effector cell infiltration, but also reverse the immunosuppressive microenvironment induced by angiogenic inducers, especially VEGF, providing rationale for the combination of anti-angiogenic therapies with immune checkpoint blockade (7-9). This issue has been addressed in our revised manuscript.

8. The text referring to fig. 1g is wrong. The cells were not rejected in the wt mice, they grew slower. This could be related to many reasons, other than rejection.

We apologize for this mistake. We have described this effect in mice as “tumor repression” instead of “tumor rejection” and have revised the text accordingly.

9. The conditioned media experiment is not clear to me as the half-life of cGAMP is thought to be minutes. It is important to measure the levels of cGAMP in the media when the cells are treated.

We thank the reviewer for raising this important point. The extracellular cGAMP levels after Cgas overexpression in Hepa1-6 cells were measured in the conditioned media, which was transferred to treat endothelial cells immediately, and a substantial increase in extracellular cGAMP levels was indeed observed (**Fig. 3d**). Extracellular cGAMP is known to be degraded by ectonucleotide pyrophosphatase/phosphodiesterase I (ENPP1) (10). However, Hepa1-6 cells had very low Enpp1 expression, as shown in the following graph. Therefore, Hepa1-6 cell-derived cGAMP is relatively stable, suitable for assessing the crosstalk between cancer cells and endothelial cells via cGAMP.

10. Fig. 4C – 5HmC is not a demethylation marker, it is an intermediate in the pathway. Once a methylated cytosine is demethylated it is no longer detectable as there is no need for ongoing demethylation. So the correlation they detected does not make sense. Moreover, the staining seems to be cytoplasmic, while TET enzymes catalyze the hydroxylation of DNA-5-methylcytosine (5mC), not free cytosine.

We thank the reviewer for the insightful comments. As the reviewer notes, correlation analysis of 5hmC with cGAS expressions in human liver cancer samples by immunohistochemical detection may not be the optimal method, and will not provide striking information in the manuscript. Therefore, we have excluded these data from

the revised manuscript.

11. The 2HG they used (disodium (R)-2-Hydroxyglutarate (Catalog No. S7873)) is not cell permeable. So how did it inhibit TET enzymes? (Fig. 5G)

We thank the reviewer for raising this critical question. As a dicarboxylic organic anion, R-2-HG is considered to be poorly cell-permeable. Nonetheless, recent studies have demonstrated that sodium-dependent dicarboxylate transporter 3 (SLC13A3) and organic anion transporters (SLC22A6 and SLC22A11) facilitate the transport of R-2-HG in several different types of cells, including renal cells, astrocytes and T cells (11). Therefore, numerous studies have utilized high concentrations of R-2-HG at millimole levels, as employed in our study, to effectively inhibit TET enzyme activity in cells and modulate cellular functions (12-15).

12. The immunostain for LRRC8C is cytoplasmic, while it should be membranous. So this is likely an artifact.

Thanks to the reviewer's question. The expression of LRRC8 family is not limited to cell membrane, LRRC8 protein expression has been shown to locate in cytoplasm such as endoplasmic reticulum (16). Moreover, both membranous and cytoplasmic expressions of LRRC8C in human HCC tissues were further confirmed in The Human Protein Atlas (HPA) database (www.proteinatlas.org), as shown in the following pictures.

13. Fig. 7k-m – looking at the graphs, it looks like there is a significant effect of both VC alone and aPDL1 alone in the CD8 depleted mice. The magnitude of this effect is similar to that in the IgG control group. There is a single outlier in the combined group,

which led them to conclude that there is no effect of the combination. I don't understand how the aPDL1 works in CD8 depleted mice.

Thanks to the reviewer's question. To more clearly assess the combined effect of VC with anti-PD-L1, we further calculated the relative tumor weight inhibition rate (**Supplementary Fig. 15e**). Although there is a single outlier in the combined group in the CD8 depleted mice, the relative tumor weight inhibition rate of combined group was significantly reduced in the CD8 depleted mice (40.3%) versus the IgG control group (76.1%). Furthermore, in addition to tumor weight, tumor volume is also important to evaluate tumor burden. As shown in **Supplementary Fig. 15c**, the magnitude of anti-tumor effect of VC and anti-PD-L1 alone or combination is remarkably diminished in the CD8 depleted mice versus the IgG control group. Therefore, these data indicate that CD8⁺ T cell mediates the combinational efficiency of VC and anti-PD-L1 therapy.

Recent studies have revealed that PD-L1 blockade enhances anti-tumor efficacy of NK cells (17,18), so it is not surprising to observe the anti-tumor effect of anti-PD-L1 in CD8 depleted mice. Nevertheless, the specific role of NK cells in VC-mediated anti-tumor immunity and anti-PD-L1 combination therapy needs further investigation.

14. Fig. 7n-p. it is clear that both aPD-L1 and VC affect growth of cGAS KO tumors; moreover, the magnitude of their effect as monodrugs, are higher in the cGAS KO tumors (particularly that of aPD-L1). Doesn't that mean that cGAS expression impedes the efficacy of ICB therapy (or at least doesn't enhance it)?

Thanks to the reviewer's question. Firstly, although we demonstrated that VC promoted anti-tumor immune response via tumor cGAS in our study, pharmacologic VC has also been shown to exert anti-tumor activity through non-immunological mechanisms such as hydrogen peroxide-induced oxidative stress to preferentially kill cancer cells (19,20). Therefore, it is not surprising that VC affect growth of cGAS KO tumors. Similarly, anti-PD-L1 probably, at least in part, affect growth of cGAS KO tumors via tumor cGAS-independent mechanism. However, the comparison the magnitude of anti-tumor effect by anti-PD-L1 between cGAS KO experiment and cGAS wt experiment is not very rigorous or accurate, because the cGAS KO experiment and cGAS wt counterpart experiments were conducted as separate, independent experiments.

Indeed, we still need to further clarify the effect of cGAS expression in liver cancer immunotherapy efficacy under our research system, despite that cGAS has been shown to potentiate immunotherapy in multiple types of tumors. Since cGAS overexpression itself developed much smaller tumors than the control cells, it is not conducive to directly compare the anti-tumor effect of immunotherapy between the two groups. To exclude that these observations resulted from an impact of tumor size on efficacy of anti-PD-L1 therapy, we adjusted cell numbers of Cgas overexpression cells and chose tumors with similar sizes to Ctrl tumors for anti-PD-L1 therapy. As expected,

compared with Ctrl group, anti-PD-L1 treatment resulted in much smaller tumors (**Supplementary Fig. 13b, c**), coincided with more intratumoral CD8⁺ cytotoxic T cells infiltration in Cgas overexpression group with similar sizes to Ctrl tumors before therapy (**Supplementary Fig. 13d**). Therefore, these findings confirm the enhanced efficacy of anti-PD-L1 therapy by cGAS expression in liver cancer regardless of tumor size.

Reviewer #3 (Remarks to the Author): with expertise in epigenetics, immunology

In this paper, Hongwei and colleagues elucidate the crosstalk between tumor and vascular endothelial cells in the tumor microenvironment via cGAS and STING. They define how the expression of cGAMP by hepatocellular carcinoma cells is used to activate STING in endothelial cells and this promotes the recruitment of T cells that

transmigrate and produce there IL-2. IL-2 activates STAT5 which recruits TET2 to specifically demethylate and promote the expression of cGAMP that again induces tumor vascular normalization. Authors propose Vitamin C as a potential immunotherapy as this can enhance the activity of TET2, improve vascular normalization and improve response to anti-PD-L1 therapy.

The study is well conducted and the results are sound. I have no additional comments and the paper is acceptable for publication.

We thank the reviewer for the positive comment.

Reviewer #4 (Remarks to the Author): with expertise in STING, cancer immunology, vascular normalization

Lv and colleagues report that tumor cell expressed TET2 epigenetically regulates intrinsic cGAS activity, resulting in local STING activation (in tumor-associated VEC leading to vascular normalization) and a proinflammatory TME conducive to effective treatment with ICI (i.e., anti-PD-L1). Vitamin C activates TET2 activity and synergizes

with anti-PD-L1 (in a tumor cGAS- and host STING-dependent manner) in effectively treating established (predominantly s.c., with 1 experiment in orthotopic) liver carcinoma models in C57BL/6 mice.

The main aspect of novelty for the current report stems from its identification of tumor cGAS as a TET2 downstream target that regulates cGAMP production/secretion and local STING activation in host cells (VEC) within the TME to regulate tumor progression, with both cGAMP export (from tumor cells) and import (by VEC) being dependent on LRRC8 channels. Others have previously shown: 1.) that tumor TET2 activity (constitutive or vitamin C-induced) mediates IFNG-JAK-STAT signaling thereby regulating, local pro-inflammatory chemokine production and PD-L1 expression, TIL recruitment and response to anti-PD-L1 in B16 melanoma and MC38 colon carcinoma models in mice while serving as a putative biomarker for predicting patient response to ICI (PMID 31310587, not cited in the current report) and that 2.) host STING activation promotes vascular normalization, improved TIL recruitment and enhanced response to ICI (including anti-PD-L1, several publications cited by the authors).

Overall, the studies appear well-performed and well-controlled, and the conclusions reached by the authors are appropriately justified. There are some areas where the report can be improved:

1.) Although the authors would like to model the impact of their operating paradigm in highly-vascularized liver cancer, the mouse model employs predominantly subcutaneous (s.c.) models with 1 instance of an orthotopic model (in Fig. 7i/j), which limits translational conclusions from being made relevant to human disease. Differences in s.c. vs. orthotopic models is not compared/contrasted, although one would expect major differences in baseline immune status between skin (pro-inflammatory) vs. liver (anti-inflammatory) that could impact mechanism of action. Inclusion of such information would improve the merit of the report.

Thanks to the reviewer's constructive comment. The major differences of immune microenvironment in s.c. vs. orthotopic models were compared by flow cytometric analysis (fig). More infiltration of immune cells including CD8⁺ T cells, NK cells, and macrophages were detected in s.c. tumor models compared to the orthotopic models (**Supplementary Fig. 14a-d**). Consequently, the therapeutic efficacy of anti-PD-L1 may be compromised in orthotopic liver cancer models. Significantly, our study demonstrated that VC treatment enhanced PD-L1 checkpoint inhibitor-induced anti-tumor responses in orthotopic liver cancer models (**Fig. 8f-h**). Moreover, as per the insightful suggestion of the reviewer, we made an exciting discovery that the addition of rIL-2 to VC treatment prominently augmented the anti-tumor efficacy of VC in combination with anti-PD-L1 therapy in orthotopic liver cancer models (**Fig. 8l-n**). Remarkably, we observed that this enhanced effect was comparable between

orthotopic and subcutaneous (s.c.) tumor models (**Fig. 8i-k**).

2.) Per the animal modeling data (Fig. 7), studies are very short term in nature (< 20 days), making it difficult to discern treatment impact on animal survival. Kaplan-Meier plots and statistical comparisons between cohorts should be provided by the authors in a revised report.

We appreciate the reviewer's comments. Due to the previous findings from our animal studies suggesting that either anti-PD-L1 monotherapy or the combination of VC and anti-PD-L1 may potentially result in complete tumor suppression after a prolonged treatment period, evaluating the impact of these treatments on animal survival is deemed inappropriate as the survival endpoint cannot be reached for both treatment groups.

Instead, we compared the endpoint time of complete tumor regression by VC and anti-PD-L1 treatment, either alone or combination over a longer period. As shown in **Supplementary Fig. 15a**, the combination therapy of VC and anti-PD-L1 exhibited a notable achievement of complete tumor regression approximately 28 days following a treatment period of 21 days. Therefore, these results indicate that VC combined with anti-PD-L1 treatment will achieve complete tumor regression after a long period of therapy.

3.) Further to the animal modeling, given the authors' indication that IL-2R/STAT5 signaling contributes to tumor cGAS activity, there is an opportunity lost in the failure to test rIL-2 along with VC +/- anti-PD-L1 for enhanced therapeutic potential in s.c. and orthotopic HCC models.

Thanks to the reviewer's constructive comment. We agree with the reviewer that this would be a valuable addition to our work and substantially enhances the overall significance of our investigation. As shown in **Fig. 8i-n**, VC treatment further boosted efficacy of anti-PD-L1 combination therapy with IL-2 in both subcutaneously implanted tumor models (**Fig. 8i-k**) and orthotopic liver cancer models (**Fig. 8l-n**). Therefore, this additional study provides a promising strategy to enhance the combinational immunotherapy for liver cancer.

4.) (minor) The report cites 80 references which seems excessive for a non-review article. The authors should reduce this number by focusing on only the most critical supportive citations.

Per the reviewer's suggestion, references have been reduced to less than 70 by removing some excessive non-review articles and meet the journal's request about references numbers (<70).

References

1. Ngo B, Van Riper JM, Cantley LC, et al. Targeting cancer vulnerabilities with high-dose vitamin C. *Nat Rev Cancer*. 2019;19(5):271-82.
2. Lan YY, Londoño D, Bouley R, Rooney MS, Hacoheh N. Dnase2a deficiency uncovers lysosomal clearance of damaged nuclear DNA via autophagy. *Cell Rep*. 2014;9(1):180-192.
3. Gehrke N, Mertens C, Zillinger T, et al. Oxidative damage of DNA confers resistance to cytosolic nuclease TREX1 degradation and potentiates STING-dependent immune sensing. *Immunity*. 2013;39(3):482-95.
4. Heidenreich R, Kappel A, Breier G. Tumor endothelium-specific transgene expression directed by vascular endothelial growth factor receptor-2 (Flk-1) promoter/enhancer sequences. *Cancer Res*. 2000;60(21):6142-7.
5. Cimmino L, Dolgalev I, Wang Y, et al. Restoration of TET2 function blocks aberrant self-renewal and leukemia progression. *Cell*. 2017;170(6):1079-95.e20.
6. Sung H, Ferlay J, Siegel RL, et al. Global Cancer Statistics 2020: GLOBOCAN Estimates of Incidence and Mortality Worldwide for 36 Cancers in 185 Countries. *CA Cancer J Clin*. 2021 May;71(3):209-249.
7. Martin JD, Seano G, Jain RK. Normalizing function of tumor vessels: progress, opportunities, and challenges. *Annu Rev Physiol*. 2019;81:505-34.
8. Tian L, Goldstein A, Wang H, et al. Mutual regulation of tumour vessel normalization and immunostimulatory reprogramming. *Nature*. 2017;544(7649):250-4.
9. Khan KA, Kerbel RS. Improving immunotherapy outcomes with anti-angiogenic treatments and vice versa. *Nat Rev Clin Oncol*. 2018;15(5):310-24.
10. Kato K, Nishimasu H, Oikawa D, et al. Structural insights into cGAMP degradation by Ecto-nucleotide pyrophosphatase phosphodiesterase 1. *Nat Commun*. 2018;9(1):4424.
11. Bunse L, Pusch S, Bunse T, et al. Suppression of antitumor T cell immunity by the oncometabolite (R)-2-hydroxyglutarate. *Nat Med*. 2018;24(8):1192-1203.
12. Zhang L, Sorensen MD, Kristensen BW, et al. D-2-Hydroxyglutarate Is an Intercellular Mediator in IDH-Mutant Gliomas Inhibiting Complement and T Cells. *Clin Cancer Res*. 2018;24(21):5381-5391.
13. Ye B, et al. Induction of functional neutrophils from mouse fibroblasts by thymidine through enhancement of Tet3 activity. *Cell Mol Immunol*. 2022;19(5):619-633.
14. Yang Q, et al. D2HGDH-mediated D2HG catabolism enhances the anti-tumor activities of CAR-T cells in an immunosuppressive microenvironment. *Mol Ther*. 2022;30(3):1188-1200.
15. Aso K, et al. Itaconate ameliorates autoimmunity by modulating T cell imbalance via metabolic and epigenetic reprogramming. *Nat Commun*. 2023;14(1):984.
16. Ghosh A, Khandelwal N, Kumar A, Bera AK. Leucine-rich repeat-containing 8B protein is associated with the endoplasmic reticulum Ca²⁺ leak in HEK293 cells. *J Cell Sci*. 2017;130(22):3818-3828.
17. Oyer JL, Gitto SB, Altomare DA, Copik AJ. PD-L1 blockade enhances anti-tumor efficacy of NK cells. *Oncoimmunology*. 2018;7(11):e1509819.
18. Dong W, Wu X, Ma S, et al. The Mechanism of Anti-PD-L1 Antibody Efficacy against PD-L1-Negative Tumors Identifies NK Cells Expressing PD-L1 as a Cytolytic Effector. *Cancer*

Discov. 2019;9(10):1422-1437.

19. Chen Q, Espey MG, Krishna MC, et al. Pharmacologic ascorbic acid concentrations selectively kill cancer cells: action as a pro-drug to deliver hydrogen peroxide to tissues. Proc Natl Acad Sci U S A. 2005;102(38):13604-9.
20. Schoenfeld JD, Sibenaller ZA, Mapuskar KA, et al. O₂^{·-} and H₂O₂-mediated disruption of Fe metabolism causes the differential susceptibility of NSCLC and GBM cancer cells to pharmacological ascorbate. Cancer Cell. 2017;31(4):487-500.e8.

REVIEWER COMMENTS

Reviewer #1 (Remarks to the Author):

The authors have improved the manuscript by enhancing the data and discussion, thereby augmenting the overall information and rigor of the paper. While there are still minor concerns, I think the paper can be accepted.

Reviewer #2 (Remarks to the Author):

The correlation between Cgas expression, extracellular cGAMP and tumor growth curves and vascular normalization is impressive and strengthens their claim. However, I do not agree with their response to the second part of my question (regarding the previous Fig. 7N). I strongly urge them to compare tumor growth between Cgas WT and KO cells, without intervention.

In their response to my second point, they provide further evidence that the blood vessels are more mature. While the data is convincing, it does not answer my concern – they didn't prove that the blood vessel normalization is a direct consequence of cGAMP on endothelial cells or blood vessels.

Re major comment 4 – then it remains unclear whether the noted effects are derived from direct effects on blood vessels, or a primary effect on any of the other players, that is then reflected in improved blood vessel maturity.

Minor comment 9 – why don't they measure cGAMP in the conditioned media experiment?

Minor comment 11 – they claim they cite numerous studies that count on the ability of cells to transport 2HG, a hydrophilic molecule. I didn't have time to go through all of them, so I looked just at the one from Nature Communications. While the authors do cite the same cat. #, in their text they state that they used a cell permeable 2HG. Most of the studies on exogenous administration of 2HG use cell permeable forms of 2HG.

Minor comment 12 – I still don't understand how an ion channel protein is cytoplasmic. The evidence brought from the human protein atlas is even more confusing as in many of the human protein atlas cases of HCC they obtain a nuclear pattern of staining (including the one from patient ID 3477 shown in the rebuttal letter). This is more likely an artifact than genuine staining of LRRC8C.

Minor comment 13 – the new data presentation (relative weight reduction), further highlights the perplexing finding, that both antiPD1 and VC, have the same effect on tumor growth regardless of the presence of CD8 T cells. This finding stands in contrast with the main claim of the manuscript. They speculate, that maybe NK cells have to do with this, but this is not shown.

Minor comment 14 – their response confuses me. Firstly, they now state that VC exerts anti-tumor activity through non-immunological mechanisms in their system. Doesn't this undermine their entire study? Second, they provide data with cGAS overexpression, while the question relates to cGAS KO cells.

Reviewer #4 (Remarks to the Author):

The authors have provided a thoughtfully revised report that addresses all of my previous concerns.

Response to reviewers' comments

Reviewer #1 (Remarks to the Author)

The authors have improved the manuscript by enhancing the data and discussion, thereby augmenting the overall information and rigor of the paper. While there are still minor concerns, I think the paper can be accepted.

We thank the reviewer for the positive comment.

Reviewer #2 (Remarks to the Author)

The correlation between Cgas expression, extracellular cGAMP and tumor growth curves and vascular normalization is impressive and strengthens their claim. However, I do not agree with their response to the second part of my question (regarding the previous Fig. 7N). I strongly urge them to compare tumor growth between Cgas WT and KO cells, without intervention.

Per the reviewer's suggestion, we compared the tumor growth between Cgas WT and KO cells without intervention. Given the initial low expression level of Cgas in native Hepa1-6 cells (**Supplementary Fig. 1a**), Cgas KO in Hepa1-6 cells exhibited a marginal augmentation of tumor growth and burdens, which did not reach statistical significance (**Supplementary Fig. 15g, h**). Nevertheless, Cgas KO in Hepa1-6, which effectively abrogates the upregulation of Cgas expression induced by VC treatment, is sufficient to confirm whether tumor vascular normalization and anti-tumor immune response prompted by VC treatment rely on cGAS expression (**Fig. 9f-h and Supplementary Fig. 12i, j**).

In their response to my second point, they provide further evidence that the blood vessels are more mature. While the data is convincing, it does not answer my concern – they didn't prove that the blood vessel normalization is a direct consequence of cGAMP on endothelial cells or blood vessels.

As shown in **Fig. 3** and **Supplementary Fig. 6**, we have revealed that tumor cells with cGAS overexpression produce cGAMP to directly activate STING pathway in endothelial cells, further suppressing angiogenesis and promoting lymphocyte trafficking via maintenance of endothelial cell junction stability and upregulation of endothelial-lymphocyte interaction-associated adhesion molecules.

Re major comment 4 – then it remains unclear whether the noted effects are derived from direct effects on blood vessels, or a primary effect on any of the other players, that is then reflected in improved blood vessel maturity.

Firstly, we have confirmed that tumor cells can directly activate endothelial STING via cGAMP (**Fig. 3**). The involvement of endothelial STING activation in vascular normalization has already been established (1,2). As mentioned by Reviewer1, this finding does not represent the primary novelty of our research investigation. The utilization of a mouse model with endothelial-specific STING knockout may elucidate the direct impact on blood vessels. Nevertheless, we believe that the considerable time

investment required for this experimental approach may not yield substantial advancements in the conceptual novelty of our research findings.

Minor comment 9 – why don't they measure cGAMP in the conditioned media experiment?

The intercellular and extracellular concentrations of cGAMP were measured in Hepa1-6-Cgas or Ctrl cells. As depicted in the right panel of **Fig. 3d**, a substantial increase in extracellular cGAMP levels was observed in the conditioned media of Hepa1-6-Cgas cells.

Minor comment 11 – they claim they cite numerous studies that count on the ability of cells to transport 2HG, a hydrophilic molecule. I didn't have time to go through all of them, so I looked just at the one from Nature Communications. While the authors do cite the same cat. #, in their text they state that they used a cell permeable 2HG. Most of the studies on exogenous administration of 2HG use cell permeable forms of 2HG. To avoid further confusion, we utilized cell-permeable forms of 2-HG (TFMB-2-HG) to replicate our previous studies and also found that TFMB-2-HG significantly decreased VC-induced Cgas expression in liver cancer cells (**Fig. 6g**).

Minor comment 12 – I still don't understand how an ion channel protein is cytoplasmic. The evidence brought from the human protein atlas is even more confusing as in many of the human protein atlas cases of HCC they obtain a nuclear pattern of staining (including the one from patient ID 3477 shown in the rebuttal letter). This is more likely an artifact than genuine staining of LRRC8C.

We appreciate the reviewer's concern but respectfully disagree with his/her reasoning. LRRC8 family proteins, functioning as ion channel proteins, are integral membrane proteins present on the plasma membrane as well as intracellular membranes such as endoplasmic reticulum (ER) and lysosomes (3,4). Hence, the detection of LRRC8 proteins in both the plasma membrane and cytoplasm is valid and rational.

Minor comment 13 – the new data presentation (relative weight reduction), further highlights the perplexing finding, that both antiPD1 and VC, have the same effect on tumor growth regardless of the presence of CD8 T cells. This finding stands in contrast with the main claim of the manuscript. They speculate, that maybe NK cells have to do with this, but this is not shown.

We conducted a comparative statistical analysis between the two experimental groups, revealing a significant reduction in the relative tumor inhibition rate of PD-L1 antibodies subsequent to the depletion of CD8⁺ T cells (**Supplementary Fig. 15e**). In contrast, VC has a similar effect on tumor growth regardless of the presence of CD8⁺ T cells. The likely explanation for this is attributed to the prominent involvement of non-immunological anti-cancer mechanisms exerted by VC.

The primary focus of our study does not center on the determination of whether the immunological or non-immunological mechanisms employed by VC exert a greater impact on its ability to suppress tumor growth. Given the constraints of our research

scope, we conducted CD8⁺ T cell clearance experiments to validate that the combinational efficiency of VC and anti-PD-L1 therapy is depending on CD8⁺ T cell-induced anti-tumor immune response (**Supplementary Fig. 15c-f**).

Minor comment 14 – their response confuses me. Firstly, they now state that VC exerts anti-tumor activity through non-immunological mechanisms in their system. Doesn't this undermine their entire study? Second, they provide data with cGAS overexpression, while the question relates to cGAS KO cells.

We apologize for the confusion. Since the non-immunological anti-tumor mechanism of VC is widely recognized, our study did not specifically address the impact of VC-induced blood vessel normalization and immune infiltration on tumor suppression. Alternatively, our research aims to elucidate the pivotal role of VC-induced blood vessel normalization in augmenting the effectiveness of immunotherapy.

Secondly, as have explained in the first-round response to the referee, we confirmed the enhanced efficacy of anti-PD-L1 therapy by cGAS overexpression in liver cancer regardless of tumor size. If necessary, to further explore whether cGAS expression impacts the efficacy of ICB therapy, we will plan to clarify the effect of cGAS expression on liver cancer immunotherapy efficacy by cGAS KO. Given the established role of tumor cGAS in enhancing immunotherapy across various tumor types (5,6), the outcomes derived from these experiments are predictable and will not contribute significantly to the conceptual novelty of our research.

Reviewer #4 (Remarks to the Author)

The authors have provided a thoughtfully revised report that addresses all of my previous concerns.

We thank the reviewer for the positive comment.

References

1. Yang H, Lee WS, Kong SJ, Kim CG, Kim JH, Chang SK, Kim S, Kim G, Chon HJ, Kim C. STING activation reprograms tumor vasculatures and synergizes with VEGFR2 blockade. *J Clin Invest.* 2019;129(10):4350-4364.
2. Campisi M, Sundararaman SK, Shelton SE, Knelson EH, Mahadevan NR, Yoshida R, Tani T, Ivanova E, Cañadas I, Osaki T, Lee SWL, Thai T, Han S, Piel BP, Gilhooley S, Paweletz CP, Chiono V, Kamm RD, Kitajima S, Barbie DA. Tumor-Derived cGAMP Regulates Activation of the Vasculature. *Front Immunol.* 2020;11:2090.
3. Ghosh A, Khandelwal N, Kumar A, Bera AK. Leucine-rich repeat-containing 8B protein is associated with the endoplasmic reticulum Ca²⁺ leak in HEK293 cells. *J Cell Sci.* 2017;130(22):3818-3828.
4. Li P, Hu M, Wang C, Feng X, Zhao Z, Yang Y, Sahoo N, Gu M, Yang Y, Xiao S, Sah R, Cover TL, Chou J, Geha R, Benavides F, Hume RI, Xu H. LRRRC8 family proteins within lysosomes regulate cellular osmoregulation and enhance cell survival to multiple physiological stresses. *Proc Natl Acad Sci U S A.* 2020;117(46):29155-

29165.

5. Kwon J, Bakhoun SF. The cytosolic DNA-sensing cGAS-STING pathway in cancer. *Cancer Discov.* 2020;10(1):26-39.
6. Jiang M, Chen P, Wang L, Li W, Chen B, Liu Y, Wang H, Zhao S, Ye L, He Y, Zhou C. cGAS-STING, an important pathway in cancer immunotherapy. *J Hematol Oncol.* 2020;13(1):81.

REVIEWERS' COMMENTS

Reviewer #2 (Remarks to the Author):

While I remain skeptical, I have no more objections.

Response to reviewers' comments

Reviewer #2 (Remarks to the Author):

While I remain skeptical, I have no more objections.

We thank the reviewer for the positive comment.